# Adaptation of olfactory receptor abundances for efficient coding

Tiberiu Teşileanu[1,2,3]*, Simona Cocco[4], Rémi Monasson[5], Vijay Balasubramanian[2,3]

[1]Center for Computational Biology, Flatiron Institute, New York, United States; [2]Initiative for the Theoretical Sciences, The Graduate Center, City University of New York, New York, United States; [3]David Rittenhouse Laboratories, University of Pennsylvania, Philadelphia, United States; [4]Laboratoire de Physique Statistique, École Normale Supérieure and CNRS UMR 8550, PSL Research, UPMC Sorbonne Université, Paris, France; [5]Laboratoire de Physique Théorique, École Normale Supérieure and CNRS UMR 8550, PSL Research, UPMC Sorbonne Université, Paris, France

**Abstract** Olfactory receptor usage is highly heterogeneous, with some receptor types being orders of magnitude more abundant than others. We propose an explanation for this striking fact: the receptor distribution is tuned to maximally represent information about the olfactory environment in a regime of efficient coding that is sensitive to the global context of correlated sensor responses. This model predicts that in mammals, where olfactory sensory neurons are replaced regularly, receptor abundances should continuously adapt to odor statistics. Experimentally, increased exposure to odorants leads variously, but reproducibly, to increased, decreased, or unchanged abundances of different activated receptors. We demonstrate that this diversity of effects is required for efficient coding when sensors are broadly correlated, and provide an algorithm for predicting which olfactory receptors should increase or decrease in abundance following specific environmental changes. Finally, we give simple dynamical rules for neural birth and death processes that might underlie this adaptation.
DOI: https://doi.org/10.7554/eLife.39279.001

**\*For correspondence:**
ttesileanu@gmail.com

**Competing interests:** The authors declare that no competing interests exist.

## Introduction

The sensory periphery acts as a gateway between the outside world and the brain, shaping what an organism can learn about its environment. This gateway has a limited capacity (*Barlow, 1961*), restricting the amount of information that can be extracted to support behavior. On the other hand, signals in the natural world typically contain many correlations that limit the unique information that is actually present in different signals. The efficient-coding hypothesis, a key normative theory of neural circuit organization, puts these two facts together, suggesting that the brain mitigates the issue of limited sensory capacity by eliminating redundancies implicit in the correlated structure of natural stimuli (*Barlow, 1961*; *van Hateren, 1992a*). This idea has led to elegant explanations of functional and circuit structure in the early visual and auditory systems (see, e.g. *Laughlin, 1981*; *Atick and Redlich, 1990*; *Van Hateren, 1993*; *Olshausen and Field, 1996*; *Simoncelli and Olshausen, 2001*; *Fairhall et al., 2001*; *Lewicki, 2002*; *Ratliff et al., 2010*; *Garrigan et al., 2010*; *Tkacik et al., 2010*; *Hermundstad et al., 2014*; *Palmer et al., 2015*; *Salisbury and Palmer, 2016*). These classic studies lacked a way to test causality by predicting how changes in the environment lead to adaptive changes in circuit composition or architecture. We propose that the olfactory system provides an avenue for such a causal test because receptor neuron populations in the mammalian nasal epithelium are regularly replaced, leading to the possibility that their abundances might adapt efficiently to the statistics of the environment.

**eLife digest** A mouse's nose contains over 10 million receptor neurons divided into about 1,000 different types, which detect airborne chemicals – called odorants – that make up smells. Each odorant activates many different receptor types. And each receptor type responds to many different odorants. To identify a smell, the brain must therefore consider the overall pattern of activation across all receptor types. Individual receptor neurons in the mammalian nose live for about 30 days, before new cells replace them. The entire population of odorant receptor neurons turns over every few weeks, even in adults.

Studies have shown that some types of these receptor neurons are used more often than others, depending on the species, and are therefore much more abundant. Moreover, the usage patterns of different receptor types can also change when individual animals are exposed to different smells. Teşileanu et al. set out to develop a computer model that can explain these observations.

The results revealed that the nose adjusts its odorant receptor neurons to provide the brain with as much information as possible about typical smells in the environment. Because each smell consists of multiple odorants, each odorant is more likely to occur alongside certain others. For example, the odorants that make up the scent of a flower are more likely to occur together than alongside the odorants in diesel. The nose takes advantage of these relationships by adjusting the abundance of the receptor types in line with them. Teşileanu et al. show that exposure to odorants leads to reproducible increases or decreases in different receptor types, depending on what would provide the brain with most information.

The number of odorant receptor neurons in the human nose decreases with time. The current findings could help scientists understand how these changes affect our sense of smell as we age. This will require collaboration between experimental and theoretical scientists to measure the odors typical of our environments, and work out how our odorant receptor neurons detect them.

DOI: https://doi.org/10.7554/eLife.39279.002

The olfactory epithelium in mammals and the antennae in insects are populated by large numbers of olfactory sensory neurons (OSNs), each of which expresses a single kind of olfactory receptor. Each type of receptor binds to many different odorants, and each odorant activates many different receptors, leading to a complex encoding of olfactory scenes (*Malnic et al., 1999*). Olfactory receptors form the largest known gene family in mammalian genomes, with hundreds to thousands of members, owing perhaps to the importance that olfaction has for an animal's fitness (*Buck and Axel, 1991*; *Tan et al., 2015*; *Chess et al., 1994*). Independently evolved large olfactory receptor families can also be found in insects (*Missbach et al., 2014*). Surprisingly, although animals possess diverse repertoires of olfactory receptors, their expression is actually highly non-uniform, with some receptors occurring much more commonly than others (*Rospars and Chambille, 1989*; *Ibarra-Soria et al., 2017*). In addition, in mammals, the olfactory epithelium experiences neural degeneration and neurogenesis, resulting in replacement of the OSNs every few weeks (*Graziadei and Graziadei, 1979*). The distribution of receptors resulting from this replacement has been found to have a mysterious dependence on olfactory experience (*Schwob et al., 1992*; *Santoro and Dulac, 2012*; *Zhao et al., 2013*; *Dias and Ressler, 2014*; *Cadiou et al., 2014*; *Ibarra-Soria et al., 2017*): increased exposure to specific ligands leads reproducibly to more receptors of some types, and no change or fewer receptors of other types.

Here, we show that these puzzling observations are predicted if the receptor distribution in the olfactory epithelium is organized to present a maximally informative picture of the odor environment. Specifically, we propose a model for the quantitative distribution of olfactory sensory neurons by receptor type. The model predicts that in a noisy odor environment: (a) the distribution of receptor types will be highly non-uniform, but reproducible given fixed receptor affinities and odor statistics; and (b) an adapting receptor neuron repertoire should reproducibly reflect changes in the olfactory environment; in a sense it should become what it smells. Precisely such findings are reported in experiments (*Schwob et al., 1992*; *Santoro and Dulac, 2012*; *Zhao et al., 2013*; *Dias and Ressler, 2014*; *Cadiou et al., 2014*; *Ibarra-Soria et al., 2017*).

In contrast to previous work applying efficient-coding ideas to the olfactory system (*Keller and Vosshall, 2007*; *McBride et al., 2014*; *Zwicker et al., 2016*; *Krishnamurthy et al., 2017*), here we take the receptor–odorant affinities to be fixed quantities and do not attempt to explain their distribution or their evolution and diversity across species. Instead, we focus on the complementary question of the optimal way in which the olfactory system can use the available receptor genes. This allows us to focus on phenomena that occur on faster timescales, such as the reorganization of the receptor repertoire as a result of neurogenesis in the mammalian epithelium.

Because of the combinatorial nature of the olfactory code (*Malnic et al., 1999*; *Stopfer et al., 2003*; *Stevens, 2015*; *Zhang and Sharpee, 2016*; *Zwicker et al., 2016*; *Krishnamurthy et al., 2017*) receptor neuron responses are highly correlated. In the absence of such correlations, efficient coding predicts that output power will be equalized across all channels if transmission limitations dominate (*Srinivasan et al., 1982*; *Olshausen and Field, 1996*; *Hermundstad et al., 2014*), or that most resources will be devoted to receptors whose responses are most variable if input noise dominates (*van Hateren, 1992a*; *Hermundstad et al., 2014*). Here, we show that the optimal solution is very different when the system of sensors is highly correlated: the adaptive change in the abundance of a particular receptor type depends critically on the global context of the correlated responses of all the receptor types in the population—we refer to this as *context-dependent adaptation*.

Correlations between the responses of olfactory receptor neurons are inevitable not only because the same odorant binds to many different receptors, but also because odors in the environment are typically composed of many different molecules, leading to correlations between the concentrations with which these odorants are encountered. Furthermore, there is no way for neural circuitry to remove these correlations in the sensory epithelium because the candidate lateral inhibition occurs downstream, in the olfactory bulb. As a result of these constraints, for an adapting receptor neuron population, our model predicts that increased activation of a given receptor type may lead to *more*, *fewer or unchanged* numbers of the receptor, but that this apparently sporadic effect will actually be reproducible between replicates. This counter-intuitive prediction matches experimental observations (*Santoro and Dulac, 2012*; *Zhao et al., 2013*; *Cadiou et al., 2014*; *Ibarra-Soria et al., 2017*).

## Olfactory response model

In vertebrates, axons from olfactory neurons converge in the olfactory bulb on compact structures called glomeruli, where they form synapses with dendrites of downstream neurons (*Hildebrand and Shepherd, 1997*); see *Figure 1a*. To good approximation, each glomerulus receives axons from only one type of OSN, and all OSNs expressing the same receptor type converge onto a small number of glomeruli, on average about two in mice to about 16 in humans (*Maresh et al., 2008*). Similar architectures can be found in insects (*Vosshall et al., 2000*).

The anatomy shows that in insects and vertebrates, olfactory information passed to the brain can be summarized by activity in the glomeruli. We treat this activity in a firing-rate approximation, which allows us to use available receptor affinity data (*Hallem and Carlson, 2006*; *Saito et al., 2009*). This approximation neglects individual spike times, which can contain important information for odor discrimination in mammals and insects (*Resulaj and Rinberg, 2015*; *DasGupta and Waddell, 2008*; *Wehr and Laurent, 1996*; *Huston et al., 2015*). Given data relating spike timing and odor exposure for different odorants and receptors, we could use the time from respiratory onset to the first elicited spike in each receptor as an indicator of activity in our model. Alternatively, we could use both the timing and the firing rate information together. Such data is not yet available for large panels of odors and receptors, and so we leave the inclusion of timing effects for future work.

A challenge specific to the study of the olfactory system as compared to other senses is the limited knowledge we have of the space of odors. It is difficult to identify common features shared by odorants that activate a given receptor type (*Rossiter, 1996*; *Malnic et al., 1999*), while attempts at defining a notion of distance in olfactory space have had only partial success (*Snitz et al., 2013*), as have attempts to find reduced-dimensionality representations of odor space (*Zarzo and Stanton, 2006*; *Koulakov et al., 2011*). In this work, we simply model the olfactory environment as a vector $\mathbf{c} = \{c_1, \ldots, c_N\}$ of concentrations, where $c_i$ is the concentration of odorant $i$ in the environment (*Figure 1a*). We note, however, that the formalism we describe here is equally applicable for other parameterizations of odor space: the components $c_i$ of the environment vector $\mathbf{c}$ could, for instance, indicate concentrations of entire classes of molecules clustered based on common chemical traits, or they might be abstract coordinates in a low-dimensional representation of olfactory space.

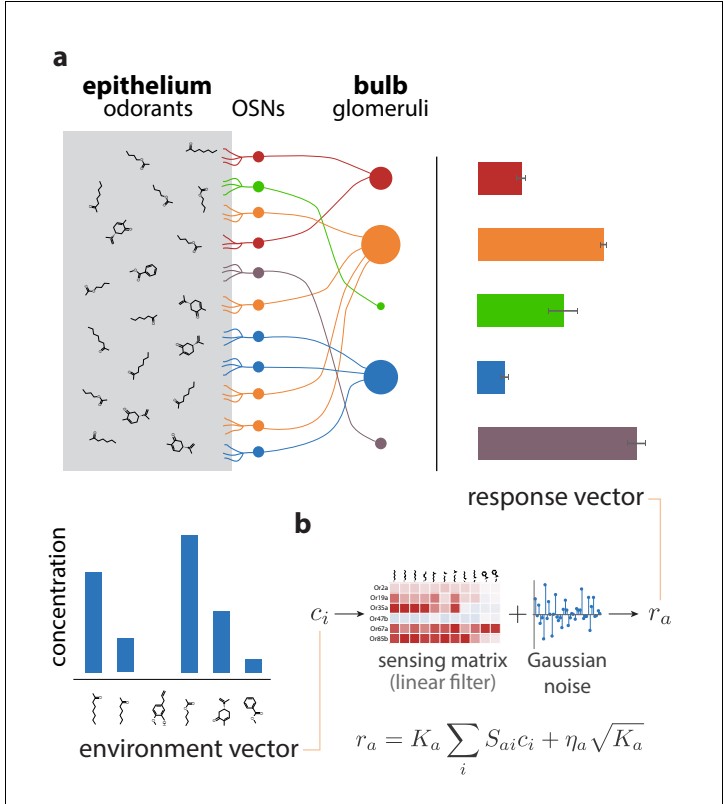

**Figure 1.** Sketch of the olfactory periphery as described in our model. (**a**) Sketch of olfactory anatomy in vertebrates. The architecture is similar in insects, with the OSNs and the glomeruli located in the antennae and antennal lobes, respectively. Different receptor types are represented by different colors in the diagram. Glomerular responses (bar plot on top right) result from mixtures of odorants in the environment (bar plot on bottom left). The response noise, shown by black error bars, depends on the number of receptor neurons of each type, illustrated in the figure by the size of the corresponding glomerulus. Glomeruli receiving input from a small number of OSNs have higher variability due to receptor noise (*e.g.*, OSN, glomerulus, and activity bar in green), while those receiving input from many OSNs have smaller variability. Response magnitudes depend also on the odorants present in the medium and the affinity profile of the receptors. (**b**) We approximate glomerular responses using a linear model based on a 'sensing matrix' *S*, perturbed by Gaussian noise $\eta_a$. $K_a$ are the numbers of OSNs of each type.

DOI: https://doi.org/10.7554/eLife.39279.003

Once a parameterization for the odor environment is chosen, we model the statistics of natural scenes by the joint probability distribution $P(c_1, \ldots, c_N)$. We are neglecting temporal correlations in olfactory cues because we are focusing on odor identity rather than olfactory search where timing of cues will be especially important. This simplifies our model, and also reduces the number of olfactory scene parameters needed as inputs. Similar static approximations of natural images have been employed powerfully along with the efficient coding hypothesis to explain diverse aspects of early vision (*e.g.*, in *Laughlin, 1981*; *Atick and Redlich, 1990*; *Olshausen and Field, 1996*; *van Hateren and van der Schaaf, 1998*; *Ratliff et al., 2010*; *Hermundstad et al., 2014*).

To construct a tractable model of the relation between natural odor statistics and olfactory receptor distributions, we describe the olfactory environment as a multivariate Gaussian with mean $\mathbf{c}_0$ and covariance matrix $\Gamma$,

$$\text{environment } P(\mathbf{c}) \sim \mathcal{N}(\mathbf{c}_0, \Gamma). \tag{1}$$

This can be thought of as a maximum-entropy approximation of the true distribution of odorant concentrations, constrained by the environmental means and covariances. This simple environmental model misses some sparse structure that is typical in olfactory scenes (*Yu et al., 2015*;

*Krishnamurthy et al., 2017*). Nevertheless, approximating natural distributions with Gaussians is common in the efficient-coding literature, and often captures enough detail to be predictive (*van Hateren, 1992a*; *van Hateren, 1992b*; *Van Hateren, 1993*; *Hermundstad et al., 2014*). This may be because early sensory systems in animals are able to adapt more effectively to low-order statistics which are easily represented by neurons in their mean activity and pairwise correlations.

The number $N$ of odorants that we use to represent an environment need not be as large as the total number of possible volatile molecules. We can instead focus on only those odorants that are likely to be encountered at meaningful concentrations by the organism that we study, leading to a much smaller value for $N$. In practice, however, we are limited by the available receptor affinity data. Our quantitative analyses are generally based on data measured using panels of 110 odorants in fly (*Hallem and Carlson, 2006*) and 63 in mammals (*Saito et al., 2009*).

We next build a model for how the activity at the glomeruli depends on the olfactory environment. We work in an approximation in which the responses depend linearly on the concentration values:

$$r_a = K_a \sum_i S_{ai} c_i + \eta_a \sqrt{K_a},$$ (2)

where $r_a$ is the response of the glomerulus indexed by $a$, $S_{ai}$ is the expected response of a single sensory neuron expressing receptor type $a$ to a unit concentration of odorant $i$, and $K_a$ is the number of neurons of type $a$. The second term describes noise, with $\eta_a$, the noise for a single OSN, modeled as a Gaussian with mean 0 and standard deviation $\sigma_a$, $\eta_a \sim \mathcal{N}(0, \sigma_a^2)$.

The approximation we are using can be seen as linearizing the responses of olfactory sensory neurons around an operating point. This has been shown to accurately capture the response of olfactory receptors to odor mixtures in certain concentration ranges (*Singh et al., 2018*). While odor concentrations in natural scenes span many orders of magnitude and are unlikely to always stay within the linear regime, the effect of the nonlinearities on the information maximization procedure that we implement below is less strong (see Appendix 3 for a comparison between our linear approximation and a nonlinear, competitive binding model in a toy example). One advantage of employing the linear approximation is that it requires a minimal set of parameters (the sensing matrix coefficients $S_{ai}$), while nonlinear models in general require additional information (such as a Hill coefficient and a maximum activation for each receptor-odorant pair for a competitive binding model; see Appendix 3).

## Information maximization

We quantify the information that responses, $\mathbf{r} = (r_1, \ldots, r_M)$, contain about the environment vector, $\mathbf{c} = (c_1, \ldots, c_N)$, using the mutual information $I(\mathbf{r}, \mathbf{c})$:

$$I(\mathbf{r}, \mathbf{c}) = \int d^M r \, d^N c \, P(\mathbf{r}, \mathbf{c}) \cdot \log \left[ \frac{P(\mathbf{r}|\mathbf{c})}{P(\mathbf{r})} \right],$$ (3)

where $P(\mathbf{r}, \mathbf{c})$ is the joint probability distribution over response and concentration vectors, $P(\mathbf{r}|\mathbf{c})$ is the distribution of responses conditioned on the environment, and $P(\mathbf{r})$ is the marginal distribution of the responses alone. Given our assumptions, all these distributions are Gaussian, and the integral can be evaluated analytically (see Appendix 2). The result is

$$I(\mathbf{r}, \mathbf{c}) = \frac{1}{2} \mathrm{Tr} \log (\mathbb{I} + \mathbb{K} \Sigma^{-1} Q),$$ (4)

where the *overlap matrix* $Q$ is related to the covariance matrix $\Gamma$ of odorant concentrations (from *Equation (1)*),

$$Q = S \Gamma S^T,$$ (5)

and $\mathbb{K}$ and $\Sigma$ are diagonal matrices of OSN abundances $K_a$ and noise variances $\sigma_a^2$, respectively:

$$\mathbb{K} = \mathrm{diag}(K_1, \ldots, K_M), \quad \Sigma = \mathrm{diag}(\sigma_1^2, \ldots, \sigma_M^2).$$ (6)

The overlap matrix $Q$ is equal to the covariance matrix of OSN responses in the absence of noise ($\sigma_a = 0$; see Appendix 2). Thus, it is a measure of the strength of the usable olfactory signal. In

contrast, the quantity $\Sigma \mathbb{K}^{-1}$ is a measure of the amount of noise in the responses, where the term $\mathbb{K}^{-1}$ corresponds to the effect of averaging over OSNs of the same type. This implies that the quantity $\mathbb{K} \Sigma^{-1} Q$ is a measure of the signal-to-noise ratio (SNR) in the system (more precisely, its square), so that *Equation (4)* represents a generalization to multiple, correlated channels of the classical result for a single Gaussian channel, $I = \frac{1}{2} \log(1 + \mathrm{SNR}^2)$ (*Shannon, 1948*; *van Hateren, 1992a*; *van Hateren, 1992b*). In the linear approximation that we are using, the information transmitted through the system is the same whether all OSNs with the same receptor type converge to one or multiple glomeruli (see Appendix 2). Because of this, for convenience we take all neurons of a given type to converge onto a single glomerulus (*Figure 1a*).

The OSN numbers $K_a$ cannot grow without bound; they are constrained by the total number of neurons in the olfactory epithelium. Thus, to find the optimal distribution of receptor types, we maximize $I(\mathbf{r}, \mathbf{c})$ with respect to $\{K_a\}$, subject to the constraints that: (1) the total number of receptor neurons is fixed ($\sum_a K_a = K_{\mathrm{tot}}$); and (2) all neuron numbers are non-negative:

$$\{K_a\} = \begin{array}{c} \arg\max \\ K_a \geq 0, \\ \sum_a K_a = K_{\mathrm{tot}} \end{array} \quad I(\mathbf{r}, \mathbf{c}). \tag{7}$$

Throughout the paper, we treat the OSN abundances $K_a$ as real numbers instead of integers, which is a good approximation as long as they are not very small. The optimization can be performed analytically using the Karush-Kuhn-Tucker (KKT) conditions (*Boyd and Vandenberghe, 2004*) (see Appendix 2), but in practice it is more convenient to use numerical optimization.

Note that in contrast to other work that has used information maximization to study the olfactory system (e.g. *Zwicker et al., 2016*), here we optimize over the OSN numbers $K_a$, while keeping the affinity profiles of the receptors (given by the sensing matrix elements $S_{ia}$) constant. Below we analyze how the optimal distribution of receptor types depends on receptor affinities, odor statistics, and the size of the olfactory epithelium.

## Receptor diversity grows with OSN population size
### Large OSN populations
In our model, receptor noise is reduced by averaging over the responses from many sensory neurons. As the number of neurons increases, $K_{\mathrm{tot}} \to \infty$, the signal-to-noise ratio (SNR) becomes very large (see *Equation (2)*). When this happens, the optimization with respect to OSN numbers $K_a$ can be solved analytically (see Appendix 2), and we find that the optimal receptor distribution is given by

$$K_a \approx K_a^{\mathrm{approx}} = \frac{K_{\mathrm{tot}}}{M} - (\sigma_a^2 A_{aa} - \overline{\sigma^2 A}), \tag{8}$$

where $A$ is the inverse of the overlap matrix $Q$ from *Equation (5)*, $A = Q^{-1}$, $\sigma_a^2$ are the receptor noise variances (*Equation (6)*), and $\overline{\sigma^2 A} = \sum \sigma_a^2 A_{aa}/M$ is a constant enforcing the constraint $\sum K_a = K_{\mathrm{tot}}$. When $K_{\mathrm{tot}}$ is sufficiently large, the constant first term dominates, meaning that the receptor distribution is essentially uniform, with each receptor type being expressed in a roughly equal fraction of the total population of sensory neurons. In this limit, the receptor distribution is as even and as diverse as possible given the genetically encoded receptor types. The small differences in abundance are related to the diagonal elements of the inverse overlap matrix $A$, modulated by the noise variances $\sigma_a^2$ (*Figure 2a*). The information maximum in this regime is shallow because only a change in OSN numbers of order $K_{\mathrm{tot}}/M$ can have a significant effect on the noise level for the activity of each glomerulus. Put another way, when the OSN numbers $K_a$ are very large, the glomerular responses are effectively noiseless, and the number of receptors of each type has little effect on the reliability of the responses. This scenario applies as long as the OSN abundances $K_a$ are much larger than the elements of the inverse overlap matrix $A$.

### Small and intermediate-sized OSN populations
When the number of neurons is very small, receptor noise can overwhelm the response to the environment. In this case, the best strategy is to focus all the available neurons on a single receptor

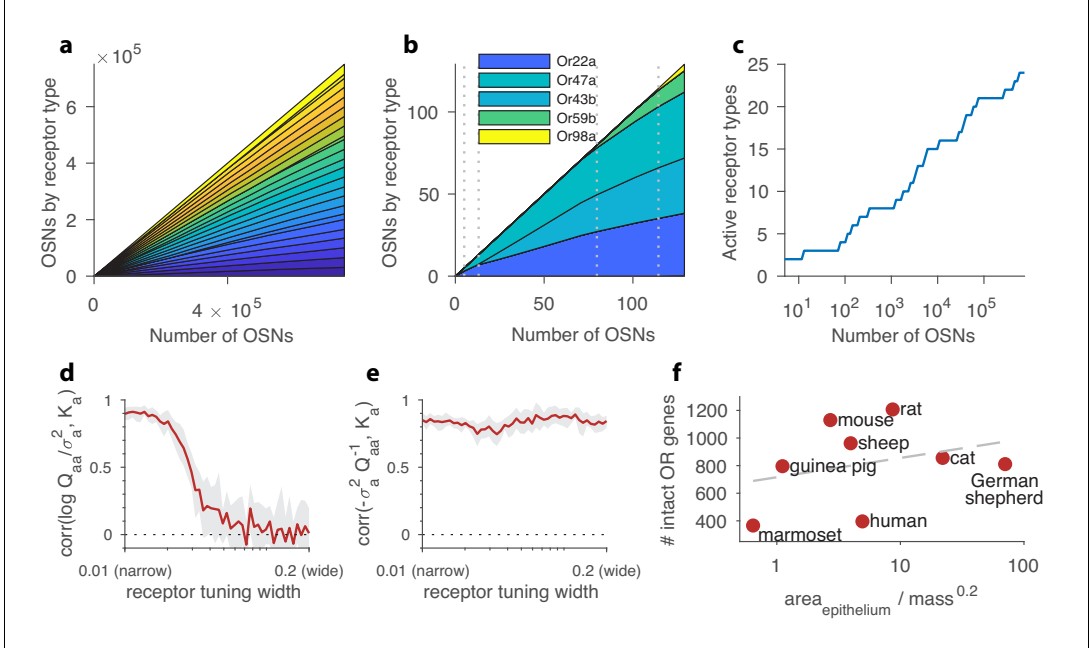

**Figure 2.** Structure of a well-adapted receptor distribution. In panels (**a–c**) the receptor sensing matrix is based on *Drosophila* (*Hallem and Carlson, 2006*) and includes 24 receptors responding to 110 odorants. In panels (**d–e**), the total number of OSNs $K_{tot}$ is fixed at 4000. In all panels, environmental odor statistics follow a random correlation matrix (see Appendix 4). Qualitative aspects are robust to variations in these choices (see Appendix 1). (**a**) Large OSN populations should have high receptor diversity (types represented by strips of different colors), and should use receptor types uniformly. (**b**) Small OSN populations should express fewer receptor types, and should use receptors non-uniformly. (**c**) New receptor types are expressed in a series of step transitions as the total number of neurons increases. Here, the odor environments and the receptor affinities are held fixed as the OSN population size is increased. (**d**) Correlation between the abundance of a given receptor type, $K_a$, and the logarithm of its signal-to-noise ratio in olfactory scenes, $\log Q_{aa}/\sigma_a^2$, shown here as a function of the tuning of the receptors. For every position along the $x$-axis, sensing matrices with a fixed receptor tuning width were generated from a random ensemble, where the tuning width indicates what fraction of all odorants elicit a strong response for the receptors (see Appendix 1). When each receptor responds strongly to only a small number of odorants, response variance is a good predictor of abundance, while this is no longer true for wide tuning. (**e**) Receptor abundances correlate well with the diagonal elements of the inverse overlap matrix normalized by the noise variances, $\sigma_a^2(Q^{-1})_{aa}$, for all tuning widths. In panels (**d–e**), the red line is the mean obtained from 24 simulations, each performed using a different sensing matrix, and the light gray area shows the interval between the 20th and 80th percentiles of results. (**f**) Number of intact olfactory receptor (OR) genes found in different species of mammals as a function of the area of the olfactory epithelium normalized to account for allometric scaling of neuron density ((*Herculano-Houzel et al., 2015*); see main text). We use this as a proxy for the number of neurons in the olfactory epithelium. Dashed line is a least-squares fit. Number of intact OR genes from (*Niimura et al., 2014*), olfactory surface area data from (*Moulton, 1967*; *Pihlström et al., 2005*; *Gross et al., 1982*; *Smith et al., 2014*), and weight data from (*Rousseeuw and Leroy, 1987*; *FCI, 2018*; *Gross et al., 1982*; *Smith et al., 2014*).

DOI: https://doi.org/10.7554/eLife.39279.004

type, thus reducing noise by summation as much as possible (*Figure 2b*). The receptor type that yields the most information will be the one whose response is most variable in natural scenes as compared to the amount of receptor noise; that is, the one that corresponds to the largest value of $Q_{aa}/\sigma_a^2$—see Appendix 2 for a derivation. This is reminiscent of a result in vision where the variance of a stimulus predicted its perceptual salience (*Hermundstad et al., 2014*).

As the total number of neurons increases, the added benefit of summing to lower noise for a single receptor type diminishes, and at some critical value it is more useful to populate a second receptor type that provides unique information not available in responses of the first type (*Figure 2b*). This process continues as the number of neurons increases, so that in an intermediate SNR range, where noise is significant but does not overwhelm the olfactory signal, our model leads to a highly non-uniform distribution of receptor types (see the trend in *Figure 2b* as the number of OSNs increases). Indeed, an inhomogeneous distribution of this kind is seen in mammals (*Ibarra-Soria et al., 2017*). Broadly, this is consistent with the idea that living systems conserve resources to

the extent possible, and thus the number of OSNs (and therefore the SNR) will be selected to be in an intermediate range in which there are just enough to make all the available receptors useful.

### Increasing OSN population size

Our model predicts that, all else being equal, the number of receptor types that are expressed should increase monotonically with the total number of sensory neurons, in a series of step transitions (see *Figure 2c*). Strictly speaking, this is a prediction that applies in a constant olfactory environment and with a fixed receptor repertoire; in terms of the parameters in our model, the total number of neurons $K_{tot}$ is varied while the sensing matrix $S$ and environmental statistics $\Gamma$ stay the same. Keeping in mind that these conditions are not usually met by distinct species, we can nevertheless ask whether, broadly speaking, there is a relation between the number of functional receptor genes and the size of the olfactory epithelium in various species.

To this end, we looked at several mammals for which the number of OR genes and the size of the olfactory epithelium were measured (*Figure 2f*). We focused on the intact OR genes (*Niimura et al., 2014*), based on the expectation that receptor genes that tend to not be used are more likely to undergo deleterious mutations. We have not found many direct measurements of the number of neurons in the epithelium for different species, so we estimated this based on the area of the olfactory epithelium (*Moulton, 1967*; *Pihlström et al., 2005*; *Gross et al., 1982*; *Smith et al., 2014*). There is a known allometric scaling relation stating that the number of neurons per unit mass for a species decreases as the 0.3 power of the typical body mass (*Herculano-Houzel et al., 2015*). Assuming a fixed number of layers in the olfactory epithelial sheet, this implies that the number of neurons in the epithelium should scale as $N_{OSN} \propto (\mathrm{epithelial\,area})/(\mathrm{body\,mass})^{\frac{2}{3} \cdot 0.3}$. We applied this relation to epithelial areas using the typical mass of several species (*Rousseeuw and Leroy, 1987*; *FCI, 2018*; *Gross et al., 1982*; *Smith et al., 2014*). The trend is consistent with expectations from our model (*Figure 2f*), keeping in mind uncertainties due to species differences in olfactory environments, receptor affinities, and behavior (e.g. consider marmoset *vs.* rat). A direct comparison is more complicated in insects, where even closely related species can vary widely in degree of specialization and thus can experience very different olfactory environments (*Dekker et al., 2006*). As we discuss below, our model's detailed predictions can be more specifically tested in controlled experiments that measure the effect of a known change in odor environment on the olfactory receptor distributions of individual mammals, as in *Ibarra-Soria et al. (2017)*.

## Optimal OSN abundances are context-dependent

We can predict the optimal distribution of receptor types given the sensing matrix $S$ and the statistics of odors by maximizing the mutual information in *Equation (4)* while keeping the total number of neurons $K_{tot} = \sum_a K_a$ constant. We tested the effect of changing the variance of a single odorant, and found that the effect on the optimal receptor abundances depends on the context of the background olfactory environment. Increased exposure to a particular ligand can lead to increased abundance of a given receptor type in one context, but to decreased abundance in another (*Figure 3*). In fact, patterns of this kind have been reported in recent experiments (*Santoro and Dulac, 2012*; *Zhao et al., 2013*; *Cadiou et al., 2014*; *Ibarra-Soria et al., 2017*). To understand this context-dependence better, we analyzed the predictions of our model in various signal and noise scenarios.

One factor that does not affect the optimal receptor distribution in our model is the average concentration vector $c_0$. This is because it corresponds to odors that are always present and therefore offer no new information about the environment. This is consistent with experiment (*Ibarra-Soria et al., 2017*), where it was observed that chronic odor exposure does not affect receptor abundances in the epithelium. In the rest of the paper, we thus restrict our attention to the covariance matrix of odorant concentrations, $\Gamma$.

The problem of maximizing the amount of information that OSN responses convey about the odor environment simplifies considerably if these responses are weakly correlated. In this case, standard efficient coding theory says that receptors whose activities fluctuate more extensively in response to the olfactory environment provide more information to brain, while receptors that are active at a constant rate or are very noisy provide less information. In this circumstance, neurons expressing receptors with large signal-to-noise ratio (SNR, i.e. signal variance as compared to noise variance) should increase in proportion relative to neurons with low signal-to-noise ratio (see

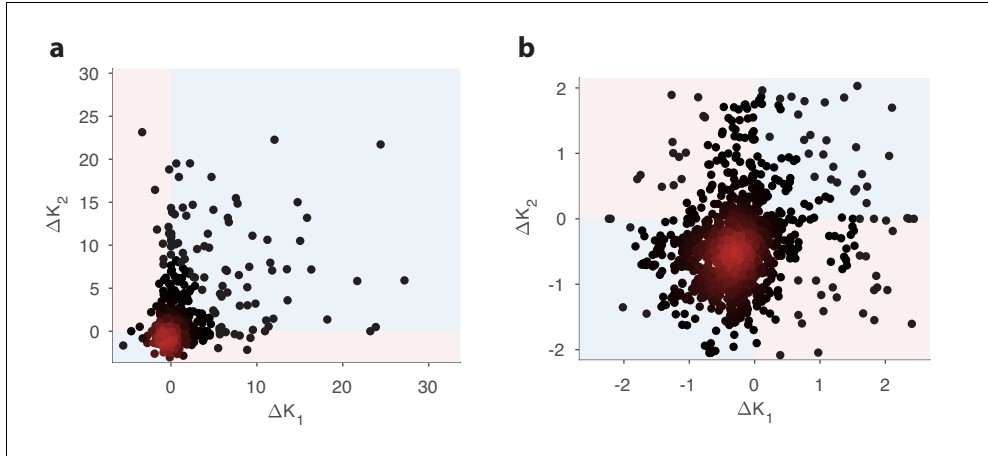

**Figure 3.** Comparison of changes in receptor abundances when the same perturbation is applied to two different environments. One hundred different pairs of environments were generated, with each environment defined by a random odor covariance matrix (procedure in Appendix 4, parameter $\beta = 8$). In each pair of environments ($i = 1, 2$), the variance of a randomly chosen odorant was increased (details in Appendix 4) to produce perturbed environments. For each receptor, we computed the optimal abundance before and after the perturbation ($K_i$ and $K_i'$) and computed the differences $\Delta K_i = K_i' - K_i$. The background environments $i = 1, 2$ in each pair set the context for the adaptive change after the perturbation. We used a sensing matrix based on fly affinity data (*Hallem and Carlson, 2006*) (24 receptors, 110 odors) and set the total OSN number to $K_{\text{tot}} = 2000$. Panel (**b**) zooms in on the central part of panel (**a**). In light blue regions, the sign of the abundance change is the same in the two contexts; light pink regions indicate opposite sign changes in the two contexts. In both figures, dark red indicates high-density regions where there are many overlapping data points.
DOI: https://doi.org/10.7554/eLife.39279.005

Appendix 2 for a derivation). In terms of our model, the signal variance of glomerular responses is given by diagonal elements of the overlap matrix $Q$ (*Equation 5*), while the noise variance is $\sigma_a^2$; so we expect $K_a$, the number of OSNs of type $a$, to increase with $Q_{aa}/\sigma_a^2$. Responses are less correlated if receptors are narrowly tuned, and we find indeed that if each receptor type responds to only a small number of odorants, the abundances of OSNs of each type correlate well with their variability in the environment (narrow-tuning side of *Figure 2d*). This is also consistent with the results at high SNR: we saw above that in that case $K_a \approx C - \sigma_a^2 (Q^{-1})_{aa}$, and when response correlations are weak, $Q$ is approximately diagonal, and thus $(Q^{-1})_{aa} \approx 1/Q_{aa}$.

The biological setting is better described in terms of widely tuned sensing matrices (*Hallem and Carlson, 2006*), and an intermediate SNR level in which noise is important, but does not dominate the responses of most receptors. We therefore generated sensing matrices with varying tuning width by changing the number of odorants that elicit strong activity in each receptor (as detailed in Appendix 1). We found that as receptors begin responding to a greater diversity of odorants, the *correlation structure* of their activity becomes important in determining the optimal receptor distribution; it is no longer sufficient to just examine the signal to noise ratios of each receptor type separately as a conventional theory suggests (wide-tuning side of *Figure 2d*). In other words, the optimal abundance of a receptor type depends not just on its activity level, but also on the context of the correlated activity levels of all the other receptor types. These correlations are determined by the covariance structures of the environment and of the sensing matrix.

In fact, across the range of tuning widths the optimal receptor abundances $K_a$ are correlated with the *inverse* of the overlap matrix, $A = Q^{-1}$ (*Figure 2e*). For narrow tuning widths, the overlap matrix $Q$ is approximately diagonal (because correlations between receptors are weak) and so $Q^{-1}$ is simply the matrix of the inverse diagonal elements of $Q$. Thus, in this limit, the correlation with $Q^{-1}$ simply follows from the correlation with $Q$ that we discussed above. As the tuning width increases keeping the total number of OSNs $K_{\text{tot}}$ constant, the responses from each receptor grow stronger, increasing the SNR, even as the off-diagonal elements of the overlap matrix $Q$ become significant. In the limit of high SNR, the analytical formula $K_a \approx C - \sigma_a^2 Q_{aa}^{-1}$ (*Equation 8*) ensures that the OSN numbers $K_a$

are still correlated with the diagonal elements of $Q^{-1}$, despite the presence of large off-diagonal components. Because of the matrix inversion in $Q^{-1}$, the optimal abundance for each receptor type is affected in this case by the full covariance structure of all the responses and not just by the variance $Q_{aa}$ of the receptor itself. Mathematically, this is because the diagonal elements of $Q^{-1}$ are functions of all the variances and covariances in the overlap matrix $Q$. This dependence of each abundance on the full covariance translates to a complex context-dependence whereby changing the same ligand in different background environments can lead to very different adapted distributions of receptors. In Appendix 6 we show that the correlation with the inverse overlap matrix has an intuitive interpretation: receptors which either do not fluctuate much or whose values can be guessed based on the responses of other receptors should have low abundances.

## Environmental changes lead to complex patterns of OSN abundance changes

To investigate how the structure of the optimal receptor repertoire varies with the olfactory environment, we first constructed a background in which the concentrations of 110 odorants were distributed according to a Gaussian with a randomly chosen covariance matrix (e.g., *Figure 4a*; see Appendix 4 for details). From this base, we generated two different environments by adding a large variance to 10 odorants in environment 1, and to 10 different odorants in environment 2

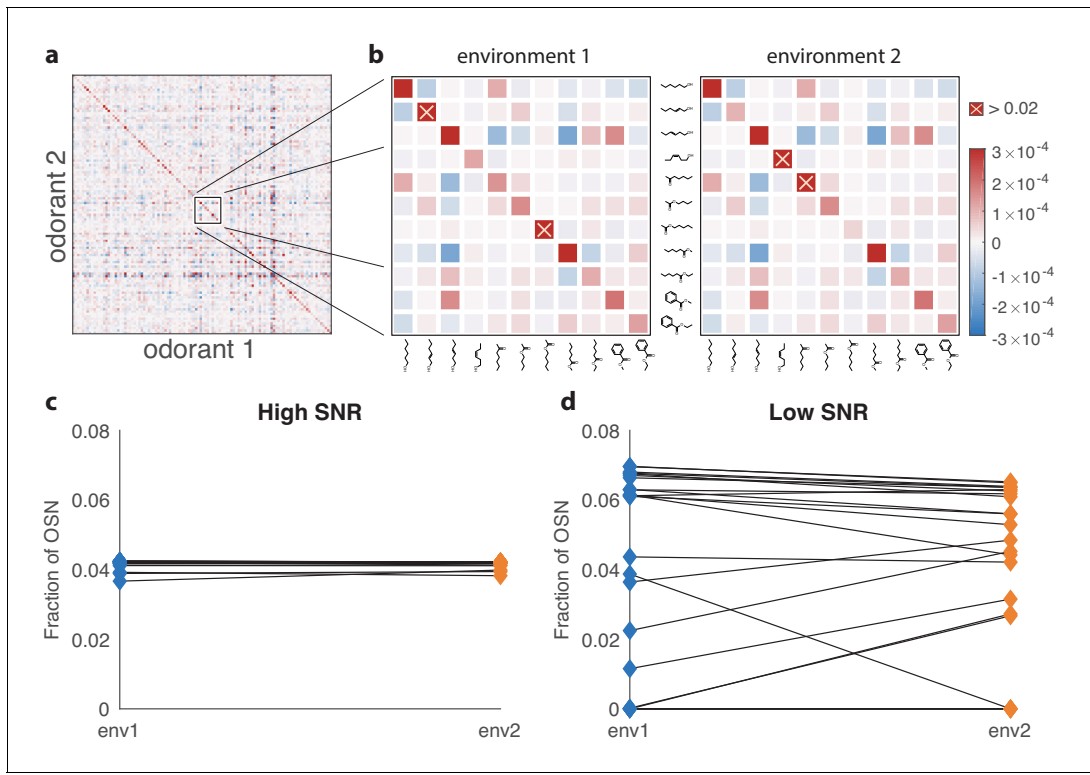

**Figure 4.** Effect of changing environment on the optimal receptor distribution. (a) An example of an environment with a random odor covariance matrix with a tunable amount of cross-correlation (details in Appendix 4). The variances are drawn from a lognormal distribution. (b) Close-ups showing some differences between the two environments used to generate results in (c and d). The two covariance matrices are obtained by adding a large variance to two different sets of 10 odorants (out of 110) in the matrix from (a). The altered odorants are identified by yellow crosses; their variances go above the color scale on the plots by a factor of more than 60. (c) Change in receptor distribution when going from environment 1 to environment 2, in conditions where the total number of receptor neurons $K_{tot}$ is large (in this case, $K_{tot} = 40\,000$), and thus the SNR is high. The blue diamonds on the left correspond to the optimal OSN fractions per receptor type in the first environment, while the orange diamonds on the right correspond to the second environment. In this high-SNR regime, the effect of the environment is small, because in both environments the optimal receptor distribution is close to uniform. (d) When the total number of neurons $K_{tot}$ is small ($K_{tot} = 100$ here) and the SNR is low, changing the environment can have a dramatic effect on optimal receptor abundances, with some receptors that are almost vanishing in one setting becoming highly abundant in the other, and vice versa.
DOI: https://doi.org/10.7554/eLife.39279.006

(*Figure 4b*). We then considered the optimal distribution in these environments for a repertoire of 24 receptor types with odor affinities inferred from (*Hallem and Carlson, 2006*). We found that when the number of olfactory sensory neurons $K_{tot}$ is large, and thus the signal-to-noise ratio is high, the change in odor statistics has little effect on the distribution of receptors (*Figure 4c*). This is because at high SNR, all the receptors are expressed nearly uniformly as discussed above, and this is true in any environment. When the number of neurons is smaller (or, equivalently, the signal-to-noise ratio is in a low or intermediate regime), the change in environment has a significant effect on the receptor distribution, with some receptor types becoming more abundant, others becoming less abundant, and yet others not changing much between the environments (see *Figure 4d*). This mimics the kinds of complex effects seen in experiments in mammals (*Schwob et al., 1992*; *Santoro and Dulac, 2012*; *Zhao et al., 2013*; *Dias and Ressler, 2014*; *Cadiou et al., 2014*; *Ibarra-Soria et al., 2017*).

## Changing odor identities has more extreme effects on receptor distributions than changing concentrations

In the comparison above, the two environment covariance matrices differed by a large amount for a small number of odors. We next compared environments with two different randomly generated covariance matrices, each generated in the same way as the background environment in *Figure 4a*. The resulting covariance matrices (*Figure 5a*, top) are very different in detail (the correlation coefficient between their entries is close to zero; distribution of changes in *Figure 5b*, red line), although they look similar by eye. Despite the large change in the detailed structure of the olfactory environment, the corresponding change in optimal receptor distribution is typically small, with a small fraction of receptor types experiencing large changes in abundance (red curve in *Figure 5c*). The average abundance of each receptor in these simulations was about 1000, and about 90% of all the abundance change values $|\Delta K_i|$ were below 20% of this, which is the range shown on the plot in *Figure 5c*. Larger changes also occurred, but very rarely: about 0.1% of the abundance changes were over 800.

If we instead engineer two environments that are almost non-overlapping so that each odorant is either common in environment 1, or in environment 2, but not in both (*Figure 5a*, bottom; see Appendix 4 for how this was done), the changes in optimal receptor abundances between environments shift away from mid-range values towards higher values (blue curve in *Figure 5c*). For instance, 40% of abundance changes lie in the range $|\Delta K| > 50$ in the non-overlapping case, while the proportion is 28% in the generic case.

It seems intuitive that animals that experience very different kinds of odors should have more striking differences in their receptor repertoires than those that merely experience the same odors with different frequencies. Intriguingly, however, our simulations suggest that the situation may be reversed at the very low end: the fraction of receptors for which the predicted abundance change is below 0.1, $|\Delta K| < 0.1$, is about 2% in the generic case but over 9% for non-overlapping environment pairs. Thus, changing between non-overlapping environments emphasizes the more extreme changes in receptor abundances, either the ones that are close to zero or the ones that are large. In contrast, a generic change in the environment leads to a more uniform distribution of abundance changes. Put differently, the particular way in which the environment changes, and not only the magnitude of the change, can affect the receptor distribution in unexpected ways.

The magnitude of the effect of environmental changes on the optimal olfactory receptor distribution is partly controlled by the tuning of the olfactory receptors (*Figure 5d*). If receptors are narrowly tuned, with each type responding to a small number of odorants, changes in the environment tend to have more drastic effects on the receptor distribution than when the receptors are broadly tuned (*Figure 5d*), an effect that could be experimentally tested.

## Model predictions qualitatively match experiments

Our study opens the exciting possibility of a causal test of the hypothesis of efficient coding in sensory systems, where a perturbation in the odor environment could lead to predictable adaptations of the olfactory receptor distribution during the lifetime of an individual. This does not happen in insects, but it can happen in mammals, since their receptor neurons regularly undergo apoptosis and are replaced.

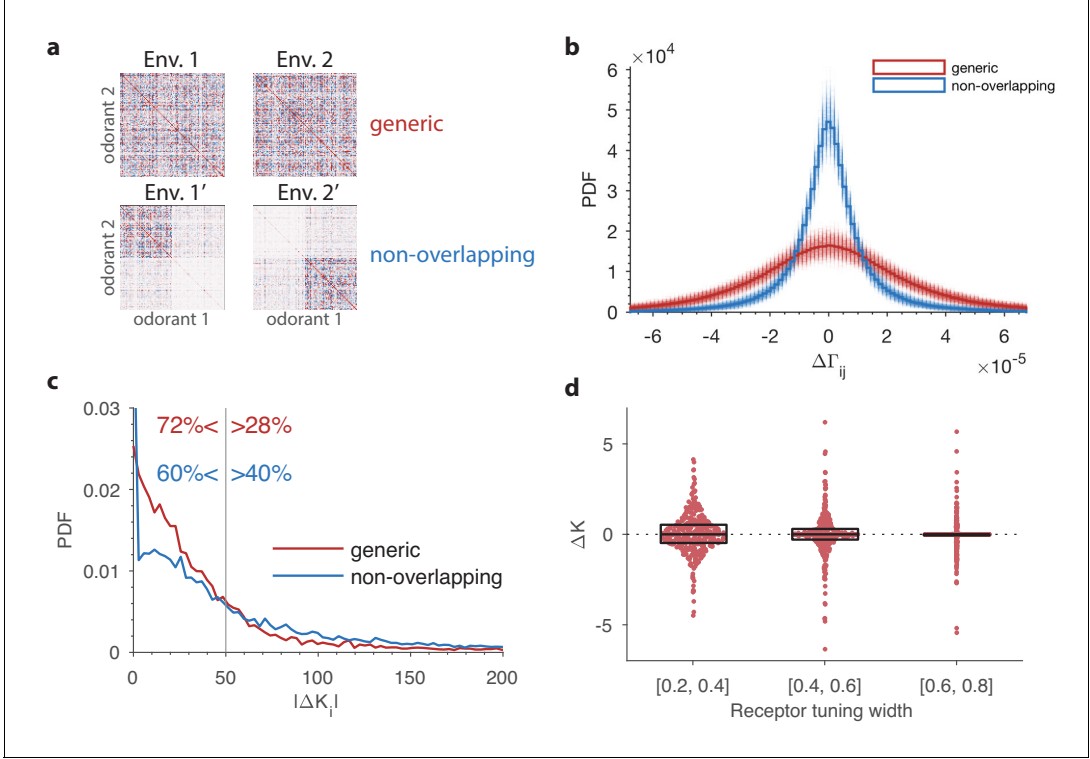

**Figure 5.** The effect of a change in environmental statistics on the optimal receptor distribution as a function of overlap in the odor content of the two environments, and the tuning properties of the olfactory receptors. (**a**) Random environment covariance matrices used in our simulations (red entries reflect positive [co-]variance; blue entries reflect negative values). The environments on the top span a similar set of odors, while those on the bottom contain largely non-overlapping sets of odors. (**b**) The distribution of changes in the elements of the environment covariance matrices between the two environments is wider (i.e. the changes tend to be larger) in the generic case than in the non-overlapping case shown in panel (**a**). The histograms in solid red and blue are obtained by pooling the 500 samples of pairs of environment matrices from each group. The plot also shows, in lighter colors, the histograms for each individual pair. (**c**) Probability distribution functions of changes in optimal OSN abundances in the 500 samples of either generic or non-overlapping environment pairs. These are obtained using receptor affinity data from the fly (**Hallem and Carlson, 2006**) with a total number of neurons $K_{tot} = 25\,000$. The non-overlapping scenario has an increased occurrence of both large changes in the OSN abundances, and small changes (the spike near the $y$-axis). The $x$-axis is cropped for clarity; the maximal values for the abundance changes $|\Delta K_i|$ are around 1000 in both cases. (**d**) Effect of tuning width on the change in OSN abundances. Here two random environment matrices obtained as in the 'generic' case from panels (**a–c**) were kept fixed, while 50 random sensing matrices with 24 receptors and 110 odorants were generated. The tuning width for each receptor, measuring the fraction of odorants that produce a significant activation of that receptor (see Appendix 1), was chosen uniformly between 0.2 and 0.8. The receptors from all the 50 trials were pooled together, sorted by their tuning width, and split into three tuning bins. Each dot represents a particular receptor in the simulations, with the vertical position indicating the amount of change in abundance $\Delta K$. The horizontal locations of the dots were randomly chosen to avoid too many overlaps; the horizontal jitter added to each point was chosen to be proportional to the probability of the observed change $\Delta K$ within its bin. This probability was determined by a kernel density estimate. The boxes show the median and interquartile range for each bin. The abundances that do not change at all ($\Delta K = 0$) are typically ones that are predicted to have zero abundance in both environments, $K_i = K'_i = 0$.

DOI: https://doi.org/10.7554/eLife.39279.007

A recent study demonstrated reproducible changes in olfactory receptor distributions of the sort that we predict in mice (**Ibarra-Soria et al., 2017**). These authors raised two groups of mice in similar conditions, exposing one group to a mixture of four odorants (acetophenone, eugenol, heptanal, and R-carvone) either continuously or intermittently (by adding the mixture to their water supply). Continuous exposure to the odorants had no effect on the receptor distribution, in agreement with the predictions of our model. In contrast, intermittent exposure did lead to systematic changes (**Figure 6a**).

We used our model to run an experiment similar to that of **Ibarra-Soria et al. (2017)** in silico (**Figure 6b**). Using a sensing matrix based on odor response curves for mouse and human receptors (data for 59 receptors from **Saito et al. (2009)**), we calculated the predicted change in OSN

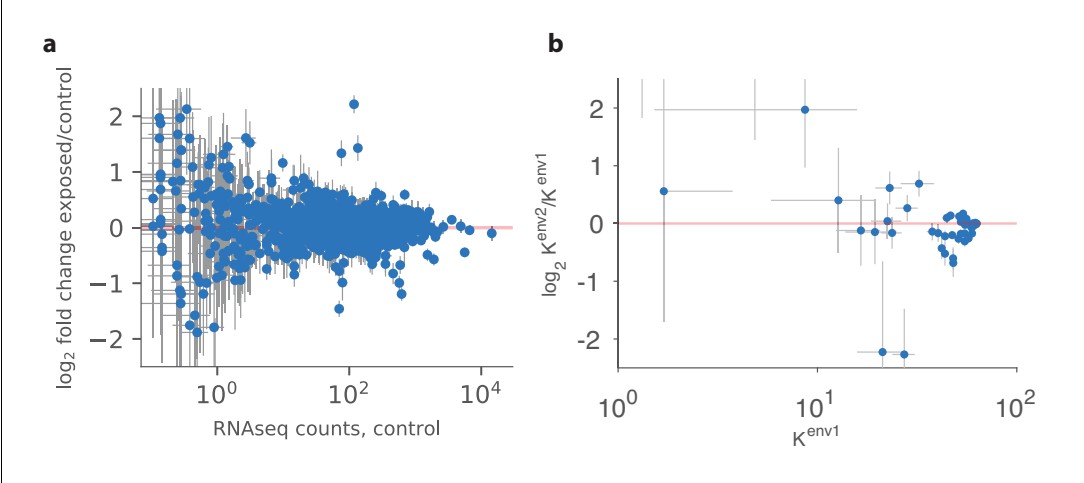

**Figure 6.** Qualitative comparison between experiment and theory. (a) Panel reproduced from raw data in *Ibarra-Soria et al. (2017)*, showing the log-ratio between receptor abundances in the mouse epithelium in the test environment (where four odorants were added to the water supply) and those in the control environment, plotted against values in control conditions (on a log scale). The error bars show standard deviation across six individuals. Compared to Figure 5B in *Ibarra-Soria et al. (2017)*, this plot does not use a Bayesian estimation technique that shrinks ratios of abundances of rare receptors toward 1 (personal communication with Professor Darren Logan, June 2017). (b) A similar plot produced in our model using mouse and human receptor response curves (*Saito et al., 2009*). The error bars show the range of variation found in the optimal receptor distribution when slightly perturbing the two environments (see the text). The simulation includes 59 receptor types for which response curves were measured (*Saito et al., 2009*), compared to 1115 receptor types assayed in *Ibarra-Soria et al. (2017)*. Our simulations used $K_{\text{tot}} = 2000$ total OSNs.

DOI: https://doi.org/10.7554/eLife.39279.008

abundances between two different environments with random covariance matrices constructed as described above. We ran the simulations 24 times, modifying the odor environments each time by adding a small amount of Gaussian random noise to the square roots of these covariance matrices to model small perturbations (details in Appendix 4; range bars in *Figure 6b*). The results show that the abundances of already numerous receptors do not change much, while there is more change for less numerous receptors. The frequencies of rare receptors can change dramatically, but are also more sensitive to perturbations of the environment (large range bars in *Figure 6b*).

These results qualitatively match experiment (*Figure 6a*), where we see the same pattern of the largest reproducible changes occurring for receptors with intermediate abundances. The experimental data is based on receptor abundance measured by RNAseq which is a proxy for counting OSN numbers (*Ibarra-Soria et al., 2017*). In our model, the distinction between receptor numbers and OSN numbers is immaterial because a change in the number of receptors expressed per neuron has the same effect as a change in neuron numbers. In general, additional experiments are needed to measure both the number of receptors per neuron and the number of neurons for each receptor type.

## A framework for a quantitative test

Given detailed information regarding the affinities of olfactory receptors, the statistics of the odor environment, and the size of the olfactory epithelium (through the total number of neurons $K_{\text{tot}}$), our model makes fully quantitative predictions for the abundances of each OSN type. Existing experiments (e.g. *Ibarra-Soria et al., 2017*) do not record necessary details regarding the odor environment of the control group and the magnitude of the perturbation experienced by the exposed group. However, such data can be collected using available experimental techniques. Anticipating future experiments, we provide a Matlab (RRID:SCR_001622) script on GitHub (RRID:SCR_002630) to calculate predicted OSN numbers from our model given experimentally-measured sensing parameters and environment covariance matrix elements (https://github.com/ttesileanu/OlfactoryReceptorDistribution).

Given the huge number of possible odorants (*Yu et al., 2015*), the sensing matrix of affinities between all receptor types in a species and all environmentally relevant odorants is difficult to

measure. One might worry that this poses a challenge for our modeling framework. One approach might be to use low-dimensional representations of olfactory space (e.g. *Koulakov et al., 2011*; *Snitz et al., 2013*), but there is not yet a consensus on the sufficiency of such representations. For now, we can ask how the predictions of our model change upon subsampling: if we only know the responses of a subset of receptors to a subset of odorants, can we still accurately predict the OSN numbers for the receptor types that we do have data for? *Figure 7a and b* show that such partial data do lead to robust statistical predictions of overall receptor abundances.

## First steps toward a dynamical model in mammals

We have explored the structure of olfactory receptor distributions that code odors efficiently, that is are adapted to maximize the amount of information that the brain gets about odors. The full solution to the optimization problem, *Equation (7)*, depends in a complicated nonlinear way on the receptor affinities $S$ and covariance of odorant concentrations $\Gamma$. The distribution of olfactory receptors in the mammalian epithelium, however, must arise dynamically from the pattern of apoptosis and neurogenesis (*Calof et al., 1996*). At a qualitative level, in the efficient coding paradigm that we propose, the receptor distribution is related to the statistics of natural odors, so that the life cycle of neurons would have to depend dynamically on olfactory experience. Such modulation of OSN lifetime by exposure to odors has been observed experimentally (*Santoro and Dulac, 2012*; *Zhao et al., 2013*) and could, for example, be mediated by feedback from the bulb (*Schwob et al., 1992*).

To obtain a dynamical model, we started with a gradient ascent algorithm for changing receptor numbers, and modified it slightly to impose the constraints that OSN numbers are non-negative, $K_a \geq 0$, and their sum $K_{\text{tot}} = \sum_a K_a$ is bounded (details in Appendix 5). This gives

$$\frac{dK_a}{dt} = \alpha \left\{ K_a - \lambda K_a^2 - \sigma_a^2 (R^{-1})_{aa} K_a^2 \right\}, \tag{9}$$

where $\alpha$ is a learning rate, $\sigma_a^2$ is the noise variance for receptor type $a$, and $R$ is the covariance matrix of glomerular responses,

$$R_{ab} = \langle r_a r_b \rangle - \langle r_a \rangle \langle r_b \rangle, \tag{10}$$

with the angle brackets denoting ensemble averaging over both odors and receptor noise. In the

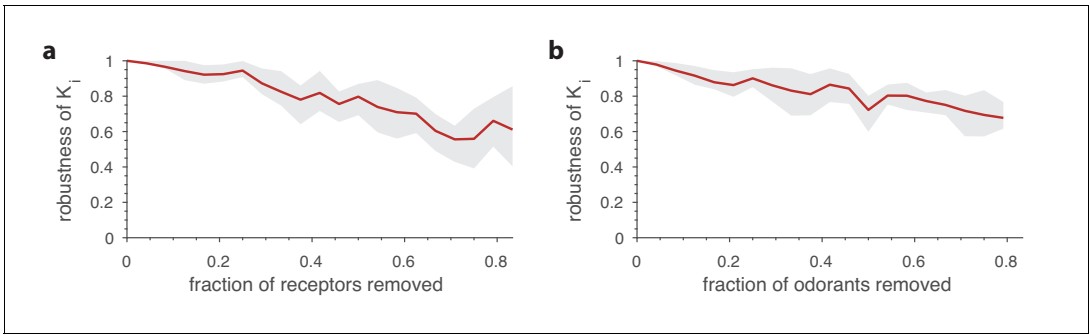

**Figure 7.** Robustness of optimal receptor distribution to subsampling of odorants and receptor types. Robustness in the prediction is measured as the Pearson correlation between the predicted OSN numbers with complete information, and after subsampling. (**a**) Robustness of OSN abundances as a function of the fraction of receptors removed from the sensing matrix. Given a full sensing matrix (in this case a 24 × 110 matrix based on *Drosophila* data (*Hallem and Carlson, 2006*)), the abundances of a subset of OSN types were calculated in two ways. First, the optimization problem from *Equation (7)* was solved including all the OSN types and an environment with a random covariance matrix (see *Figure 5*). Then a second optimization problem was run in which a fraction of the OSN types were removed. The optimal neuron counts $K_i'$ obtained using the second method were then compared (using the Pearson correlation coefficient) against the corresponding numbers $K_i$ from the full optimization. The shaded area in the plot shows the range between the 20th and 80th percentiles for the correlation values obtained in 10 trials, while the red curve is the mean. A new subset of receptors to be removed and a new environment covariance matrix were generated for each sample. (**b**) Robustness of OSN abundances as a function of the fraction of odorants removed from the environment, calculated similarly to panel a except now a certain fraction of odorants was removed from the environment covariance matrix, and from the corresponding columns of the sensing matrix.
DOI: https://doi.org/10.7554/eLife.39279.009

absence of the experience-related term $(R^{-1})_{aa}$, the dynamics from *Equation (9)* would be simply logistic growth: the population of OSNs of type $a$ would initially grow at a rate $\alpha$, but would saturate when $K_a = 1/\lambda$ because of the population-dependent death rate $\lambda K_a$. In other words, the quantity $M/\lambda$ sets the asymptotic value for the total population of sensory neurons, $K_{\text{tot}} \to M/\lambda$, with $M$ being the number of receptor types.

Because of the last term in *Equation (9)*, the death rate in our model is influenced by olfactory experience in a receptor-dependent way. In contrast, the birth rate is not experience-dependent and is the same for all OSN types. Indeed, in experiments, the odor environment is seen to have little effect on receptor choice, but does modulate the rate of apoptosis in the olfactory epithelium (*Santoro and Dulac, 2012*). Our results suggest that, if olfactory sensory neuron lifetimes are appropriately anti-correlated with the inverse response covariance matrix, then the receptor distribution in the epithelium can converge to achieve optimal information transfer to the brain.

The elements of the response covariance matrix $R_{ab}$ could be estimated by temporal averaging of co-occurring glomerular activations via lateral connections between glomeruli (*Mori et al., 1999*). Performing the inverse necessary for our model is more intricate. The computations could perhaps be done by circuits in the bulb and then fed back to the epithelium through known mechanisms (*Schwob et al., 1992*),

Within our model, *Figure 8a* shows an example of receptor numbers converging to the optimum from random initial values. The sensing matrix used here is based on mammalian data (*Saito et al., 2009*) and we set the total OSN number to $K_{\text{tot}} = 2000$. The environment covariance matrix is generated using the random procedure described earlier (details in Appendix 4). We see that some receptor types take longer than others to converge (the time axis is logarithmic, which helps visualize the whole range of convergence behaviors). Roughly speaking, convergence is slower when the final OSN abundance is small, which is related to the fact that the rate of change $dK_a/dt$ in *Equation (9)* vanishes in the limit $K_a \to 0$. For the same reason, OSN populations that start at a very low level also take a long time to converge.

In *Figure 8b*, we show convergence to the same final state, but this time starting from a distribution that is not random but was optimized for a different environment. The initial and final environments are the same as the two environments used in the previous section to compare the simulations to experimental findings (*Figure 6b*). Interestingly, many receptor types actually take longer to converge in this case compared to the random starting point, perhaps because there are local optima in the landscape of receptor distributions. Given such local minima, stochastic fluctuations will allow the dynamics to reach the global optimum more easily. In realistic situations, there are many sources of such variability, for example, sampling noise due to the fact that the response covariance matrix $R$ must be estimated through stochastic odor encounters and noisy receptor

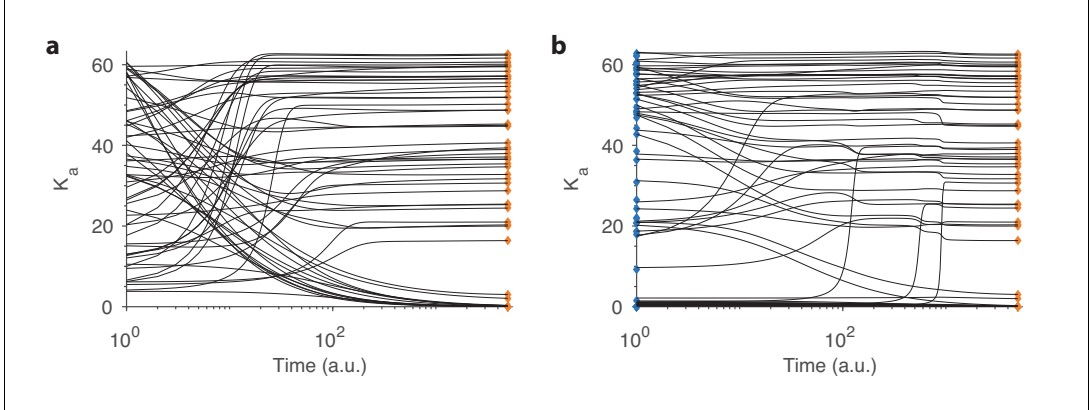

**Figure 8.** Convergence in our dynamical model. (a) Example convergence curves in our dynamical model showing how the optimal receptor distribution (orange diamonds) is reached from a random initial distribution of receptors. Note that the time axis is logarithmic. (b) Convergence curves when starting close to the optimal distribution from one environment (blue diamonds) but optimizing for another. A small, random deviation from the optimal receptor abundance in the initial environment was added (see text).
DOI: https://doi.org/10.7554/eLife.39279.010

readings. In fact, in *Figure 8b*, we added a small amount of noise (corresponding to $\pm 0.05 K_{\text{tot}}/M$) to the initial distribution of receptors to improve convergence rates.

## Discussion

We built a model for the distribution of receptor types in the olfactory epithelium that is based on efficient coding, and assumes that the abundances of different receptor types are adapted to the statistics of natural odors in a way that maximizes the amount of information conveyed to the brain by glomerular responses. This model predicts a non-uniform distribution of receptor types in the olfactory epithelium, as well as reproducible changes in the receptor distribution after perturbations to the odor environment. In contrast to other applications of efficient coding, our model operates in a regime in which there are significant correlations between sensors because the adaptation of OSN abundances occurs upstream of the brain circuitry that can decorrelate olfactory responses. In this regime, OSN abundances depend on the full correlation structure of the inputs, leading to predictions that are context-dependent in the sense that whether the abundance of a specific receptor type goes up or down due to a shift in the environment depends on the global context of the responses of all the other receptors. All these striking phenomena have been observed in recent experiments and had not been explained prior to this study.

In our framework, the sensitivity of the receptor distribution to changes in odor statistics is affected by the tuning of the olfactory receptors, with narrowly tuned receptors being more readily affected by such changes than broadly tuned ones. The model also predicts that environments that differ in the identity of the odors that are present will lead to greater deviations in the optimal receptor distribution than environments that differ only in the statistics with which these odors are encountered. Likewise, the model broadly predicts a monotonic relationship between the number of receptor types found in the epithelium and the total number of olfactory sensory neurons, all else being equal.

A detailed test of our model requires more comprehensive measurements of olfactory environments than are currently available. Our hope is that studies such as ours will spur interest in measuring the natural statistics of odors, opening the door for a variety of theoretical advances in olfaction, similar to what was done for vision and audition. Such measurements could for instance be performed by using mass spectrometry to measure the chemical composition of typical odor scenes. Given such data, and a library of receptor affinities, our GitHub (RRID:SCR_002630) online repository provides an easy-to-use script that uses our model to predict OSN abundances. For mammals, controlled changes in environments similar to those in *Ibarra-Soria et al. (2017)* could provide an even more stringent test for our framework.

To our knowledge, this is the first time that efficient coding ideas have been used to explain the pattern of usage of receptors in the olfactory epithelium. Our work can be extended in several ways. OSN responses can manifest complex, nonlinear responses to odor mixtures. Accurate models for how neurons in the olfactory epithelium respond to complex mixtures of odorants are just starting to be developed (e.g. *Singh et al., 2018*), and these can in principle be incorporated in an information-maximization procedure similar to ours. More realistic descriptions of natural odor environments can also be added, as they amount to changing the environmental distribution $P(\mathbf{c})$. For example, the distribution of odorants could be modeled using a Gaussian mixture, rather than the normal distribution used in this paper to enable analytic calculations. Each Gaussian in the mixture would model a different odor object in the environment, more closely approximating the sparse nature of olfactory scenes discussed in, for example, *Krishnamurthy et al. (2017)*.

Of course, the goal of the olfactory system is not simply to encode odors in a way that is optimal for decoding the concentrations of volatile molecules in the environment, but rather to provide an encoding that is most useful for guiding future behavior. This means that the value of different odors might be an important component shaping the neural circuits of the olfactory system. In applications of efficient coding to vision and audition, maximizing mutual information, as we did, has proved effective even in the absence of a treatment of value (*Laughlin, 1981*; *Atick and Redlich, 1990*; *van Hateren, 1992a*; *Olshausen and Field, 1996*; *Simoncelli and Olshausen, 2001*; *Fairhall et al., 2001*; *Lewicki, 2002*; *Ratliff et al., 2010*; *Garrigan et al., 2010*; *Tkacik et al., 2010*; *Hermundstad et al., 2014*; *Palmer et al., 2015*; *Salisbury and Palmer, 2016*). However, in general, understanding the role of value in shaping neural circuits is an important experimental and

theoretical problem. To extend our model in this direction, we would replace the mutual information between odorant concentrations and glomerular responses by a different function that takes into account value assignments (see, e.g. *Rivoire and Leibler, 2011*). It could be argued, though, that such specialization to the most behaviorally relevant stimuli might be unnecessary or even counter-productive close to the sensory periphery. Indeed, a highly specialized olfactory system might be better at reacting to known stimuli, but would be vulnerable to adversarial attacks in which other organisms take advantage of blind spots in coverage. Because of this, and because precise information regarding how different animals assign value to different odors is scarce, we leave these considerations for future work.

One exciting possibility suggested by our model is a way to perform a first causal test of the efficient coding hypothesis for sensory coding. Given sufficiently detailed information regarding receptor affinities and natural odor statistics, experiments could be designed that perturb the environment in specified ways, and then measure the change in olfactory receptor distributions. Comparing the results to the changes predicted by our theory would provide a strong test of efficient coding by early sensory systems in the brain.

## Materials and methods

### Software and data

The code (written in Matlab, RRID:SCR_001622) and data that we used to generate all the results and figures in the paper is available on GitHub (RRID:SCR_002630), at https://github.com/ttesileanu/OlfactoryReceptorDistribution (*Teşileanu, 2019*; copy archived at https://github.com/elifesciences-publications/OlfactoryReceptorDistribution).

## Acknowledgements

We thank Joel Mainland and David Zwicker for helpful discussions, and Elissa Hallem, Joel Mainland, and Darren Logan for olfactory receptor affinity data. This work was supported by a grant from the Simons Foundation/SFARI Mathematical Modeling in Living Systems program (400425, VB). VB was also supported by Aspen Center for Physics NSF grant PHY-160761 and US–Israel Binational Science Foundation grant 2011058. TT was supported by the Swartz Foundation. This work was also supported by NSF grant PHY-1734030 (Center for the Physics of Biological Function).

## Additional information

### Funding

| Funder | Grant reference number | Author |
|---|---|---|
| Simons Foundation | 400425 | Vijay Balasubramanian |
| Aspen Center for Physics | PHY-160761 | Vijay Balasubramanian |
| Swartz Foundation | | Tiberiu Teşileanu |
| National Science Foundation | PHY-1734030 | Tiberiu Teşileanu<br>Vijay Balasubramanian |
| United States - Israel Binational Science Foundation | 2011058 | Vijay Balasubramanian |

The funders had no role in study design, data collection and interpretation, or the decision to submit the work for publication.

### Author contributions

Tiberiu Teşileanu, Conceptualization, Software, Formal analysis, Validation, Visualization, Methodology, Writing—original draft, Writing—review and editing; Simona Cocco, Vijay Balasubramanian, Conceptualization, Formal analysis, Supervision, Methodology, Writing—review and editing; Rémi

Monasson, Conceptualization, Software, Formal analysis, Supervision, Methodology, Writing—review and editing

## Author ORCIDs
Tiberiu Teşileanu (iD) http://orcid.org/0000-0003-3107-3088
Rémi Monasson (iD) https://orcid.org/0000-0002-4459-0204
Vijay Balasubramanian (iD) https://orcid.org/0000-0002-6497-3819

## Decision letter and Author response
Decision letter https://doi.org/10.7554/eLife.39279.024
Author response https://doi.org/10.7554/eLife.39279.025

## Additional files

### Supplementary files
• Transparent reporting form
DOI: https://doi.org/10.7554/eLife.39279.011

### Data availability
All the code necessary to reproduce our results and the figures from the paper is available on GitHub, at https://github.com/ttesileanu/OlfactoryReceptorDistribution (copy archived at https://github.com/elifesciences-publications/OlfactoryReceptorDistribution). The olfactory receptor affinity data were originally published in Hallem et al. (2006) and Saito et al. (2009), and the olfactory receptor expression levels in mouse were originally published in Ibarra-Soria et al. (2017).

The following datasets were generated:

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

## Appendix 1

DOI: https://doi.org/10.7554/eLife.39279.012

# Choice of sensing matrices and receptor noise variances

We used three types of sensing matrices in this study. Two were based on experimental data, one using fly receptors (*Hallem and Carlson, 2006*), and one using mouse and human receptors (*Saito et al., 2009*); and another type of sensing matrix was based on randomly-generated receptor affinity profiles. These can all be either directly downloaded from our repository on GitHub (RRID:SCR_002630), https://github.com/ttesileanu/OlfactoryReceptorDistribution, or generated using the code available there.

## Fly sensing matrix

Some of our simulations used a sensing matrix based on *Drosophila* receptor affinities, as measured by Hallem and Carlson (*Hallem and Carlson, 2006*). This includes the responses of 24 of the 60 receptor types in the fly against a panel of 110 odorants, measured using single-unit electrophysiology in a mutant antennal neuron. We used the values from Table S1 in (*Hallem and Carlson, 2006*) for the sensing matrix elements. To estimate receptor noise, we used the standard deviation measured for the background firing rates for each receptor (data obtained from the authors). The fly data has the advantage of being more complete than equivalent datasets in mammals.

## Mammalian sensing matrix

When comparing our model to experimental findings from (*Ibarra-Soria et al., 2017*), we used a sensing matrix based on mouse and human receptor affinity data from (*Saito et al., 2009*). This was measured using heterologous expression of olfactory genes, and tested in total 219 mouse and 245 human receptor types against 93 different odorants. However, only 49 mouse receptors and 10 human receptors exhibited detectable responses against any of the odorants, while only 63 odorants activated any receptors. From the remaining $59 \times 63 = 3717$ receptor–odorant pairs, only 335 (about 9%) showed a response, and were assayed at 11 different concentration points. In this paper, we used the values obtained for the highest concentration (3 mM).

## Random sensing matrices

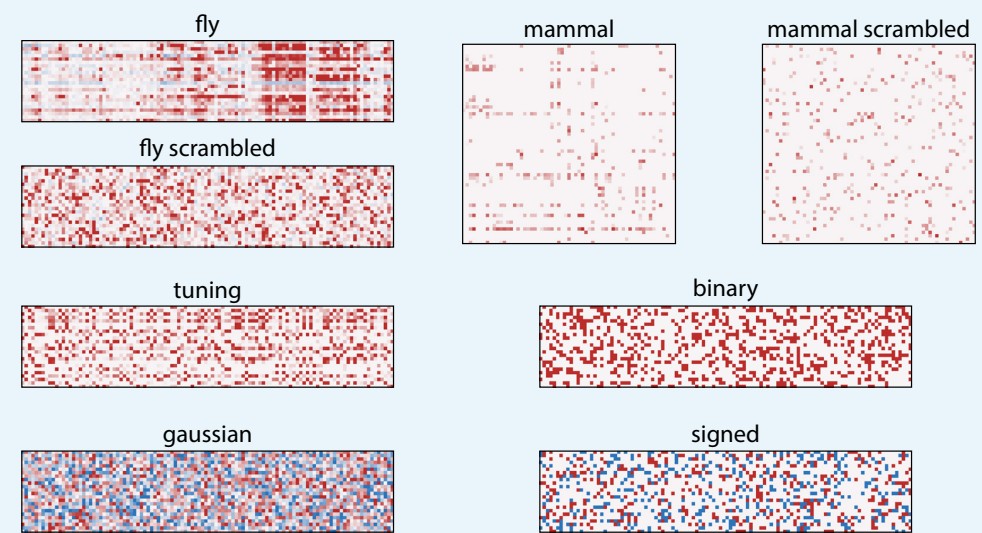

**Appendix 1—figure 1.** Heat maps of the types of sensing matrices used in our study. The color scaling is arbitrary, with red representing positive values and blue negative values. 'Fly' and 'mammal' are the sensing matrices based on *Drosophila* receptor affinities (**Hallem and Carlson, 2006**), and mouse and human affinities (**Saito et al., 2009**), respectively. 'Fly scrambled' and 'mammal scrambled' are permutations of the 'fly' and 'mammal' matrices in which elements are arbitrarily scrambled. 'Tuning', 'gaussian', 'binary', and 'signed' are random sensing matrix generated as described in the *Random sensing matrices* section.
DOI: https://doi.org/10.7554/eLife.39279.013

The random sensing matrices matrices used in the main text (and referred to as 'tuning' in some of the figures in this Appendix) were generated as follows. We started by treating the column (i.e. odorant) index as a one-dimensional odor coordinate with periodic boundary conditions. We normalized the index to a coordinate $x$ running from 0 to 1. For each receptor, we then chose a center $x_0$ along this line, corresponding to the odorant to which the receptor has maximum affinity, and a standard deviation $\sigma$, corresponding to the tuning width of the receptor. Note that both $x_0$ and $\sigma$ are allowed to be real numbers, so that the maximum affinity can occur at a position that does not correspond to any particular odorant from the sensing matrix.

To obtain a bell-like response profile for the receptors while preserving the periodicity of the odor coordinate we chose, we defined the response affinity to odorant $x$ by

$$\phi(x) = \exp\left[-\frac{1}{2}\left(\frac{2\sin\pi(x-x_0)}{\sigma}\right)^2\right]. \tag{11}$$

This expression can be obtained by imagining odorant space as a circle embedded in a two-dimensional plane, with odorant $x$ mapped to an angle $\theta = 2\pi x$ on this circle, and considering a Gaussian response profile in this two-dimensional embedding space. This is simply a convenient choice for treating odor space in a way that eliminates artifacts at the edges of the sensing matrix, and we do not assign any significance to the particular coordinate system that we used.

The centers $x_0$ for the Gaussian profiles for each of the receptors were chosen uniformly at random, and the tuning width $\sigma$ was either a fixed parameter for the entire sensing matrix, or was uniformly sampled from an interval. Before using the matrices we randomly shuffled the columns to remove the dependencies between neighboring odorants, and finally added some amount of random Gaussian noise (mean centered and with standard deviation 1/200). The overall scale of the sensing matrices was set by multiplying all the affinities by 100, which yielded values comparable to the measured firing rates in fly olfactory neurons (**Hallem and Carlson, 2006**).

For the robustness results below we also generated random matrices in additional ways: (1) 'gaussian': drawing the affinities from a Gaussian distribution (with zero mean and standard deviation 2), (2) 'bernoulli': drawing from a Bernoulli distribution (with elements equal to 5 with probability 30%, and 0 with probability 70%), (3) 'signed': drawing from a Bernoulli distribution followed by choosing the sign (so that elements are 5 with probability 15%, –5 with probability 15%, and 0 with probability 70%); and (4, 5) 'fly scrambled' and 'mammal scrambled': scrambling the elements in the fly and mammalian datasets (across both odorants and receptors).

## Robustness of results to changing the sensing matrix

Our qualitative results are robust across a variety of different choices for the sensing matrix (*Appendix 1—figure 1*). For instance, the optimal number of receptor types expressed in a fraction of the OSN population larger than 1% grows monotonically with the total number of neurons (*Appendix 1—figure 2*). Similarly, the general effect that environment change has on optimal OSN numbers, with less abundant receptor types changing more than more abundant ones, is generic across different choices of sensing matrices (*Appendix 1—figure 3*).

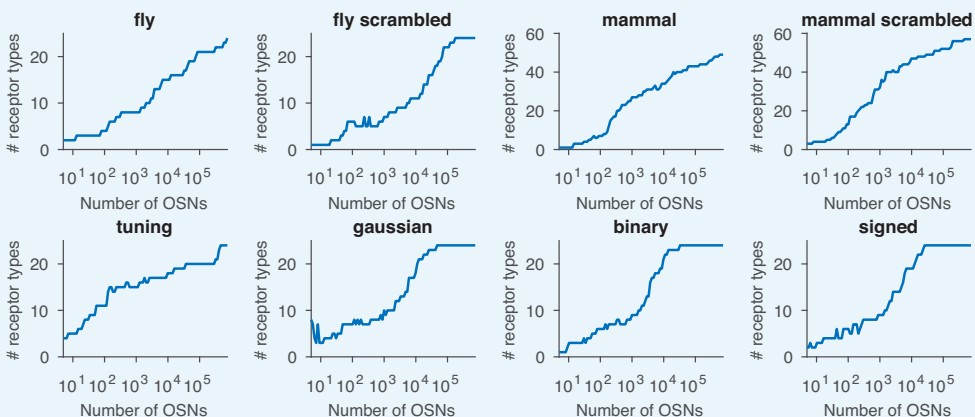

**Appendix 1—figure 2.** Effect of sensing matrix on the dependence between the number of receptor types expressed in the optimal distribution and the total number of OSNs. The labels refer to the sensing matrices from *Appendix 1—figure 1*.
DOI: https://doi.org/10.7554/eLife.39279.014

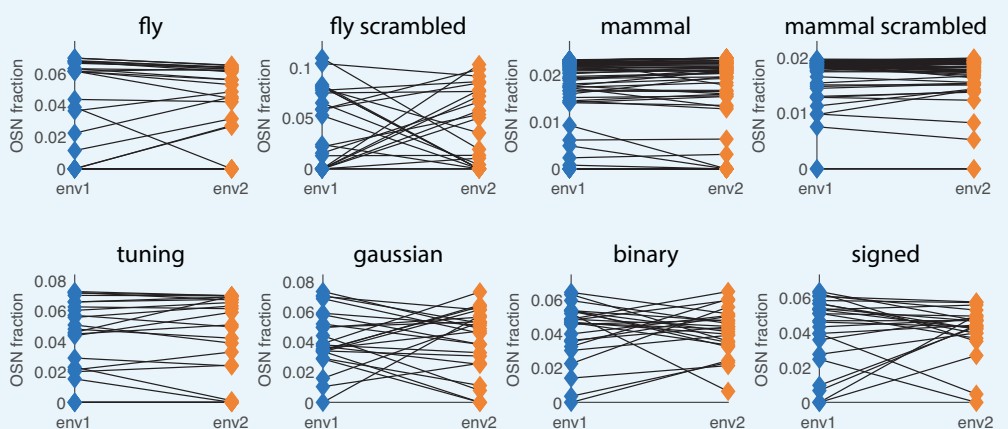

**Appendix 1—figure 3.** Different choices of sensing matrix lead to similar behavior of optimal receptor distribution under environment change. The labels refer to the sensing matrices from *Appendix 1—figure 1*, whose scales were adjusted to ensure that the simulations are in a low

SNR regime. The blue (orange) diamonds on the left (right) side of each plot represent the optimal OSN abundances in environment 1 (environment 2). The two environment covariance matrices are obtained by starting with a background randomly-generated covariance matrix (as described below) and adding a large amount of variance to two different sets of 10 odorants (out of 110 for most sensing matrices, and 63 for the 'mouse' and 'mouse scrambled' ones).

DOI: https://doi.org/10.7554/eLife.39279.015

## Appendix 2

DOI: https://doi.org/10.7554/eLife.39279.012

## Mathematical derivations

### Deriving the expression for the mutual information

In the main text we assume a Gaussian distribution for odorant concentrations and approximate receptor responses as linear with additive Gaussian noise, *Equation (2)*. Thus it follows that the marginal distribution of receptor responses is also Gaussian. Taking averages of the responses, $\langle r_a \rangle$, and of products of responses, $\langle r_a r_b \rangle$, over both the noise distribution and the odorant distribution, and using *Equation (2)* from the main text, we get a normal distribution of responses:

$$\mathbf{r} \sim \mathcal{N}(\mathbf{r}_0, R), \tag{12}$$

where the mean response vector $\mathbf{r}_0$ and the response covariance matrix $R$ are given by

$$\begin{aligned} \mathbf{r}_0 &= \mathbb{K}S\mathbf{c}_0, \\ R &= [\Sigma + \mathbb{K}Q]\mathbb{K}, \end{aligned} \tag{13}$$

where $S$ is the sensing matrix, $\mathbb{K}$ is a diagonal matrix of OSN abundances, and $\Sigma$ is the covariance matrix of receptor noises, $\Sigma = \mathrm{diag}(\sigma_1^2, \ldots, \sigma_M^2)$ (also see the main text). Here, as in *Equation (1)* in the main text, $\mathbf{c}_0$ is the mean concentration vector, $\Gamma$ is the covariance matrix of odorant concentrations, and we use the overlap matrix from *Equation (5)* in the main text, $Q = S\Gamma S^T$. Note that in the absence of noise ($\Sigma = 0$), the response matrix is simply the overlap matrix $Q$ modulated by the number of OSNs of each type, $R_{\mathrm{noiseless}} = \mathbb{K}Q\mathbb{K}$.

The joint probability distribution over responses and concentrations, $P(\mathbf{r}, \mathbf{c})$, is itself Gaussian. To calculate the corresponding covariance matrix, we need the covariances between responses, $\langle r_a r_b \rangle - \langle r_a \rangle \langle r_b \rangle$, which are just the elements of the response matrix $R$ from *Equation (13)* above; and between concentrations, $\langle c_i c_j \rangle - \langle c_i \rangle \langle c_j \rangle$, which are the elements of the environment covariance matrix $\Gamma$, *Equation (1)* in the main text. In addition, we need the covariances between responses and concentrations, $\langle r_a c_i \rangle - \langle r_a \rangle \langle c_i \rangle$, which can be calculated using *Equation (2)* from the main text. We get:

$$(\mathbf{r}, \mathbf{c}) \sim \mathcal{N}((\mathbf{r}_0, \mathbf{c}_0), \Lambda), \tag{14}$$

with

$$\Lambda = \begin{pmatrix} R & \mathbb{K}S\Gamma \\ \Gamma S^T \mathbb{K} & \Gamma \end{pmatrix}. \tag{15}$$

The mutual information between responses and odors is then given by (see below for a derivation):

$$I(\mathbf{r}, \mathbf{c}) = \frac{1}{2} \log \frac{\det \Gamma \det R}{\det \Lambda}. \tag{16}$$

From *Equation (13)* we have

$$\det R = \det(\Sigma + \mathbb{K}Q) \det \mathbb{K}, \tag{17}$$

and from *Equation (15)*,

$$\begin{aligned}
\det\Lambda &= \det\begin{pmatrix} R & \mathbb{K}S\Gamma \\ \Gamma S^T\mathbb{K} & \Gamma \end{pmatrix} = \det\Gamma \cdot \det(R - \mathbb{K}S\Gamma\Gamma^{-1}\Gamma S^T\mathbb{K}) \\
&= \det\Gamma \cdot \det(\Sigma\mathbb{K} + \mathbb{K}Q\mathbb{K} - \mathbb{K}S\Gamma S^T\mathbb{K}) \\
&= \det\Gamma \cdot \det\Sigma\mathbb{K},
\end{aligned}$$
(18)

where we used **Equation (13)** again, and employed Schur's determinant identity (derived below). Thus,

$$I(\mathbf{r},\mathbf{c}) = \frac{1}{2}\log\frac{\det\Gamma \cdot \det(\Sigma + \mathbb{K}Q) \cdot \det\mathbb{K}}{\det\Gamma\det\Sigma\det\mathbb{K}} = \frac{1}{2}\log\det(\mathbb{I} + \Sigma^{-1}\mathbb{K}Q)$$
(19)

This recovers the result quoted in the main text, **Equation (4)**.

By using the fact that the diagonal matrices $\mathbb{K}$ and $\Sigma^{-1}$ commute, we can also write:

$$\begin{aligned}
I(\mathbf{r},\mathbf{c}) &= \frac{1}{2}\log\det(\Sigma^{-1/2}\Sigma^{1/2} + \Sigma^{-1}\mathbb{K}Q) = \frac{1}{2}\log\det\Sigma^{-1/2}(\Sigma^{1/2} + \Sigma^{-1/2}\mathbb{K}Q) \\
&= \frac{1}{2}\log\det(\Sigma^{1/2} + \Sigma^{-1/2}\mathbb{K}Q)\Sigma^{-1/2} = \frac{1}{2}\log\det(\mathbb{I} + \mathbb{K}\Sigma^{-1/2}Q\Sigma^{-1/2}) \\
&= \frac{1}{2}\log\det(\mathbb{I} + \mathbb{K}\tilde{Q}).
\end{aligned}$$
(20)

This shows that the mutual information can be written in terms of a symmetric 'SNR matrix' $\tilde{Q} = \Sigma^{-1/2}Q\Sigma^{-1/2}$. This is simply the covariance matrix of responses in which each response was normalized by the noise variance of the corresponding receptor.

## Schur's determinant identity

The identity for the determinant of a $2 \times 2$ block matrix that we used in **Equation (18)** above can be derived in the following way. First, note that

$$\begin{pmatrix} A & B \\ C & D \end{pmatrix} = \begin{pmatrix} \mathbb{I} & B \\ 0 & D \end{pmatrix}\begin{pmatrix} A - BD^{-1}C & 0 \\ D^{-1}C & \mathbb{I} \end{pmatrix}.$$
(21)

Now, from the definition of the determinant it can be seen that

$$\det\begin{pmatrix} A & B \\ 0 & \mathbb{I} \end{pmatrix} = \det\begin{pmatrix} A & 0 \\ C & \mathbb{I} \end{pmatrix} = \det A,$$
(22)

since all the products involving elements from the off-diagonal blocks must necessarily also involve elements from the 0 matrix. Thus, taking the determinant of **Equation (21)**, we get the desired identity

$$\det\begin{pmatrix} A & B \\ C & D \end{pmatrix} = \det D \cdot \det(A - BD^{-1}C).$$
(23)

## Mutual information for Gaussian distributions

The expression from **Equation (16)** for the mutual information $I(\mathbf{r},\mathbf{c})$ can be derived by starting with the fact that $I$ is equal to the Kullback-Leibler (KL) divergence from the joint distribution $P(\mathbf{r},\mathbf{c})$ to the product distribution $P(\mathbf{r})P(\mathbf{c})$. As a first step, let us calculate the KL divergence between two multivariate normals in $n$ dimensions:

$$D = D_{\mathrm{KL}}(p||q) = \int p(\mathbf{x})\log\frac{p(\mathbf{x})}{q(\mathbf{x})}\,d\mathbf{x},$$
(24)

where

$$p(\mathbf{x}) = \frac{1}{\sqrt{(2\pi)^n \det A}} \exp\left[-\frac{1}{2}(\mathbf{x}-\mu_A)^T A^{-1}(\mathbf{x}-\mu_A)\right],$$

$$q(\mathbf{x}) = \frac{1}{\sqrt{(2\pi)^n \det B}} \exp\left[-\frac{1}{2}(\mathbf{x}-\mu_B)^T B^{-1}(\mathbf{x}-\mu_B)\right]. \tag{25}$$

Plugging the distribution functions into the logarithm, we have

$$D = \frac{1}{2}\log\frac{\det B}{\det A} + \frac{1}{2}\int p(\mathbf{x})\left[(\mathbf{x}-\mu_B)^T B^{-1}(\mathbf{x}-\mu_B) - (\mathbf{x}-\mu_A)^T A^{-1}(\mathbf{x}-\mu_A)\right]d\mathbf{x}, \tag{26}$$

where the normalization property of $p(\mathbf{x})$ was used. Using also the definition of the mean and of the covariance matrix, we have

$$\int p(\mathbf{x}) x_i \, d\mathbf{x} = \mu_{A,i}, \tag{27a}$$

$$\int p(\mathbf{x}) x_i x_j \, d\mathbf{x} = A_{ij}, \tag{27b}$$

which implies

$$\int p(\mathbf{x})(\mathbf{x}-\mu)^T C^{-1}(\mathbf{x}-\mu) \, d\mathbf{x} = \mathrm{Tr}(AC^{-1}) + (\mu_A - \mu)^T C^{-1}(\mu_A - \mu) \tag{28}$$

for any vector $\mu$ and matrix $C$. Plugging this into *Equation (26)*, we get

$$D = \frac{1}{2}\log\frac{\det B}{\det A} + \frac{1}{2}\left[\mathrm{Tr}(AB^{-1}) - n\right] + \frac{1}{2}(\mu_A - \mu_B)^T B^{-1}(\mu_A - \mu_B). \tag{29}$$

We can now return to calculating the KL divergence from $P(\mathbf{r},\mathbf{c})$ to $P(\mathbf{r})P(\mathbf{c})$. Note that, since $P(\mathbf{r})$ and $P(\mathbf{c})$ are just the marginals of the joint distribution, the means of the variables are the same in the joint and in the product, so that the last term in the KL divergence vanishes. The covariance matrix for the product distribution is

$$\Lambda_{\mathrm{prod}} = \begin{pmatrix} R & 0 \\ 0 & \Gamma \end{pmatrix}, \tag{30}$$

so the product inside the trace becomes

$$\Lambda\Lambda_{\mathrm{prod}}^{-1} = \begin{pmatrix} R & \cdots \\ \cdots & \Gamma \end{pmatrix}\begin{pmatrix} R^{-1} & 0 \\ 0 & \Gamma^{-1} \end{pmatrix} = \begin{pmatrix} \mathbb{I} & \cdots \\ \cdots & \mathbb{I} \end{pmatrix}, \tag{31}$$

where the entries replaced by '$\cdots$' need not be calculated because they drop out when the trace is taken. The sum of the dimensions of $R$ and $\Gamma$ is equal to the dimension, $n$, of $\Lambda$, so that the term involving the trace from *Equation (29)* also drops out, leaving us with the final result:

$$I = D_{\mathrm{KL}}(p(\mathbf{r},\mathbf{c})p(\mathbf{r})p(\mathbf{c})) = \frac{1}{2}\log\frac{\det R \det \Gamma}{\det \Lambda}, \tag{32}$$

which is the same as *Equation (16)* that was used in the previous section.

## Deriving the KKT conditions for the information optimum

In order to find the optimal distribution of olfactory receptors, we must maximize the mutual information from *Equation (4)* in the main text, subject to constraints. Let us first calculate the gradient of the mutual information with respect to the receptor numbers:

$$\frac{\partial I}{\partial K_a} = \frac{1}{2}\frac{\partial}{\partial K_a}\log\det(\mathbb{I} + \mathbb{K}\tilde{Q}) = \frac{1}{2}\frac{\partial}{\partial K_a}\mathrm{Tr}\log(\mathbb{I} + \mathbb{K}\tilde{Q}). \tag{33}$$

The cyclic property of the trace allows us to use the usual rules to differentiate under the trace operator, so we get

$$\frac{\partial I}{\partial K_a} = \frac{1}{2}\mathrm{Tr}\left[\frac{\partial \mathbb{K}}{\partial K_a}\left(\tilde{Q}^{-1}+\mathbb{K}\right)^{-1}\right] = \frac{1}{2}\sum_{b,c}\frac{\partial(K_b\delta_{bc})}{\partial K_a}\left(\tilde{Q}^{-1}+\mathbb{K}\right)^{-1}_{ca}$$

$$= \frac{1}{2}\left(\tilde{Q}^{-1}+\mathbb{K}\right)^{-1}_{aa}. \tag{34}$$

We now have to address the constraints. We have two kinds of constraints: an equality constraint that sets the total number of neurons, $\sum K_a = K_{\mathrm{tot}}$; and inequality constraints that ensure that all receptor abundances are non-negative, $K_a \geq 0$. This can be done using the Karush-Kuhn-Tucker (KKT) conditions, which require the introduction of Lagrange multipliers: $\lambda$ for the equality constraint, and $\mu_a$ for the inequality constraints. At the optimum, we must have:

$$\frac{\partial I}{\partial K_a} = \frac{1}{2}\lambda\frac{\partial}{\partial K_a}\left(\sum_b K_b - K_{\mathrm{tot}}\right) - \sum_b \mu_b\frac{\partial}{\partial K_a}K_b$$

$$= \lambda - \mu_a, \tag{35}$$

where the Lagrange multipliers for the inequality constraints, $\mu_a$, must be non-negative, and must vanish unless the inequality is saturated:

$$\begin{aligned}\mu_a &\geq 0,\\ \mu_a K_a &= 0.\end{aligned} \tag{36}$$

Put differently, if $K_a > 0$, then $\mu_a = 0$ and $\partial I/\partial K_a = \lambda/2$; while if $K_a = 0$, then $\partial I/\partial K_a = \lambda/2 - \mu_a \leq \lambda/2$. Combined with **Equation (34)**, this yields

$$\begin{cases}\left(\tilde{Q}^{-1}+\mathbb{K}\right)^{-1}_{aa} = \lambda, & \text{if } K_a > 0, \text{ or}\\ \left(\tilde{Q}^{-1}+\mathbb{K}\right)^{-1}_{aa} < \lambda, & \text{if } K_a = 0.\end{cases} \tag{37}$$

The magnitude of $\lambda$ is set by imposing the normalization condition $\sum K_a = K_{\mathrm{tot}}$.

## The many-neuron approximation

Suppose we are in the regime in which the total number of neurons is large, and in particular, each of the abundances $K_a$ is large. Then we can perform an expansion of the expression appearing in the KKT equations from **Equation (37)**:

$$\left(\tilde{Q}^{-1}+\mathbb{K}\right)^{-1} = \mathbb{K}^{-1}\left(\mathbb{I}+\tilde{Q}^{-1}\mathbb{K}^{-1}\right)^{-1} \approx \mathbb{K}^{-1}\left(\mathbb{I}-\tilde{Q}^{-1}\mathbb{K}^{-1}\right), \tag{38}$$

whose $aa$ component is

$$\left(\tilde{Q}^{-1}+\mathbb{K}\right)^{-1}_{aa} \approx \frac{1}{K_a}\left[1-\frac{\tilde{Q}^{-1}_{aa}}{K_a}\right] = \frac{1}{K_a}\left[1-\frac{\sigma_a^2 Q^{-1}_{aa}}{K_a}\right], \tag{39}$$

where we used $\tilde{Q} = \Sigma^{-1/2}Q\Sigma^{1/2}$. With the notation

$$A = Q^{-1}, \tag{40}$$

we can plug into **Equation (37)** and get

$$\lambda \approx \frac{1}{K_a} - \frac{\sigma_a^2 A_{aa}}{K_a^2}. \tag{41}$$

This quadratic equation has only one large solution, and it is given approximately by

$$K_a \approx \frac{1}{\lambda} - \sigma_a^2 A_{aa}. \tag{42}$$

Combined with the normalization constraint, $\sum_a K_a = K_{\mathrm{tot}}$, this recovers **Equation (8)** from the main text.

## Optimal distribution for uncorrelated responses

When the overlap matrix $Q = S\Gamma S^T$ is diagonal, the optimization problem simplifies considerably. By plugging $Q = diag(Q_{aa})$ into **Equation (4)** in the main text, we find

$$
\begin{aligned}
I(\mathbf{r}, \mathbf{c}) &= \frac{1}{2}\log\det(\mathbb{I} + \Sigma^{-1}\mathbb{K}Q) = \frac{1}{2}\log\det \operatorname{diag}(1 + K_a Q_{aa}/\sigma_a^2) \\
&= \frac{1}{2}\sum_a \log\left(1 + K_a \frac{Q_{aa}}{\sigma_a^2}\right).
\end{aligned}
\tag{43}
$$

We can again use the KKT approach and add Lagrange multipliers $\lambda$ and $\mu_a$ for enforcing the equality and inequality constraints, respectively,

$$
\bar{I} = \frac{1}{2}\sum_a \log\left(1 + K_a \frac{Q_{aa}}{\sigma_a^2}\right) - \lambda \sum_a K_a - \mu_a K_a,
\tag{44}
$$

and take derivatives with respect to $K_a$ to find the optimum,

$$
0 = \frac{\partial \bar{I}}{\partial K_a} = \frac{1}{2}\frac{1}{K_a + \sigma_a^2/Q_{aa}} - \lambda - \mu_a,
\tag{45}
$$

with the condition that $\mu_a \geq 0$ and either $\mu_a$ or $K_a$ must vanish, $\mu_a K_a = 0$. This leads to

$$
K_a = \max\left(0, \frac{1}{2\lambda} - \frac{\sigma_a^2}{Q_{aa}}\right),
\tag{46}
$$

showing that receptor abundances grow monotonically with $Q_{aa}/\sigma_a^2$. This explains the correlation between OSN abundances $K_a$ and receptor SNRs $Q_{aa}/\sigma_a^2$ when the responses are uncorrelated or weakly correlated.

## First receptor type to be activated

When there is only one active receptor, $K_x = K_{\text{tot}}$, $K_{a \neq x} = 0$, the KKT conditions from **Equation (37)** are automatically satisfied. The receptor that is activated first can be found in this case by calculating the information $I(\mathbf{r}, \mathbf{c})$ using **Equation (4)** from the main text while assuming an arbitrary index $x$ for the active receptor, and then finding $x = x^*$ that yields the maximum value. Without loss of generality, we can permute the receptor indices such that $x = 1$. Using **Equation (19)** and setting $K_1 = K_{\text{tot}}$, we have:

$$
\begin{aligned}
I_1(\mathbf{r}, \mathbf{c}) &= \frac{1}{2}\operatorname{Tr}\log(\mathbb{I} + \mathbb{K}\Sigma^{-1}Q) = \frac{1}{2}\log\det(\mathbb{I} + \mathbb{K}\Sigma^{-1}Q) \\
&= \frac{1}{2}\log\begin{vmatrix} 1 + K_{\text{tot}}Q_{11}/\sigma_1^2 & K_{\text{tot}}Q_{12}/\sigma_1^2 & \cdots & K_{\text{tot}}Q_{1M}/\sigma_1^2 \\ 0 & 1 & & 0 \\ \vdots & & \ddots & \\ 0 & 0 & \cdots & 1 \end{vmatrix} \\
&= \frac{1}{2}\log(1 + \frac{K_{\text{tot}}Q_{11}}{\sigma_1^2}).
\end{aligned}
\tag{47}
$$

Thus, in general, the information when only receptor type $x$ is activated is given by

$$
I_x(\mathbf{r}, \mathbf{c}) = \frac{1}{2}\log(1 + \frac{K_{\text{tot}}Q_{xx}}{\sigma_x^2}),
\tag{48}
$$

which implies that information is maximized when $x$ matches the receptor corresponding to the highest ratio between the diagonal value of the overlap matrix $Q$ and the receptor variance in that channel $\sigma_x^2$; that is the receptor that maximizes the signal-to-noise ratio:

$$x^* = \arg\max \frac{Q_{xx}}{\sigma_x^2} = \arg\max \tilde{Q}_{xx} \equiv \arg\max \text{SNR}_x. \tag{49}$$

Another way to think of this result is by employing the usual expression for the capacity of a single Gaussian channel, and then finding the channel that maximizes this capacity.

## Invariance of mutual information under invertible and differentiable transformations

Consider the mutual information between two variables $\mathbf{r} \in \mathbb{R}^M$ and $\mathbf{c} \in \mathbb{R}^N$:

$$I(\mathbf{r}, \mathbf{c}) = \int d^M r \, d^N c \, P(\mathbf{r}, \mathbf{c}) \cdot \log\left[\frac{P(\mathbf{r}|\mathbf{c})}{P(\mathbf{r})}\right]. \tag{50}$$

Let us now define two different variables that depend on $\mathbf{r}$ and $\mathbf{c}$ in an invertible and continuously-differentiable (but in general nonlinear) way,

$$\mathbf{y} = \mathbf{y}(\mathbf{r}), \quad \mathbf{x} = \mathbf{x}(\mathbf{c}). \tag{51}$$

The joint probability distribution for the new variables is related to the joint distribution of the original variables through the Jacobian determinants,

$$P(\mathbf{y}, \mathbf{x}) = P(\mathbf{r}, \mathbf{c}) \det \mathbb{J}_r \det \mathbb{J}_c, \tag{52}$$

where

$$\mathbb{J}_r = \begin{pmatrix} \frac{\partial r_1}{\partial y_1} & \cdots & \frac{\partial r_1}{\partial y_M} \\ \vdots & \ddots & \vdots \\ \frac{\partial r_M}{\partial y_1} & \cdots & \frac{\partial r_M}{\partial y_M} \end{pmatrix}, \quad \mathbb{J}_c = \begin{pmatrix} \frac{\partial c_1}{\partial x_1} & \cdots & \frac{\partial c_1}{\partial x_N} \\ \vdots & \ddots & \vdots \\ \frac{\partial c_N}{\partial x_1} & \cdots & \frac{\partial c_N}{\partial x_N} \end{pmatrix}. \tag{53}$$

For the marginals, we have

$$\begin{aligned} P(\mathbf{y}) &= \int d^N x \, P(\mathbf{y}, \mathbf{x}) = \int d^N c \, \frac{1}{\det \mathbb{J}_c} P(\mathbf{r}, \mathbf{c}) \det \mathbb{J}_r \det \mathbb{J}_c = P(\mathbf{r}) \det \mathbb{J}_r, \\ P(\mathbf{x}) &= \int d^M y \, P(\mathbf{y}, \mathbf{x}) = \int d^M r \, \frac{1}{\det \mathbb{J}_r} P(\mathbf{r}, \mathbf{c}) \det \mathbb{J}_r \det \mathbb{J}_c = P(\mathbf{c}) \det \mathbb{J}_c, \end{aligned} \tag{54}$$

where we used the standard substitution formula for multiple integrals. We can now calculate the mutual information between the new variables:

$$\begin{aligned} I(\mathbf{y}, \mathbf{x}) &= \int d^M y \, d^N x \, P(\mathbf{y}, \mathbf{x}) \cdot \log\left[\frac{P(\mathbf{y}|\mathbf{x})}{P(\mathbf{y})}\right] = \int d^M y \, d^N x \, P(\mathbf{y}, \mathbf{x}) \cdot \log\left[\frac{P(\mathbf{y}, \mathbf{x})}{P(\mathbf{y})P(\mathbf{x})}\right] \\ &= \int d^M r \, d^N c \, \frac{1}{\det \mathbb{J}_r} \frac{1}{\det \mathbb{J}_c} P(\mathbf{r}, \mathbf{c}) \det \mathbb{J}_r \det \mathbb{J}_c \cdot \log\left[\frac{P(\mathbf{r}, \mathbf{c}) \det \mathbb{J}_r \det \mathbb{J}_c}{P(\mathbf{r}) \det \mathbb{J}_r P(\mathbf{c}) \det \mathbb{J}_c}\right] \\ &= \int d^M r \, d^N c \, P(\mathbf{r}, \mathbf{c}) \cdot \log\left[\frac{P(\mathbf{r}, \mathbf{c})}{P(\mathbf{r})P(\mathbf{c})}\right] \\ &\equiv I(\mathbf{r}, \mathbf{c}). \end{aligned} \tag{55}$$

Thus, invertible and continuously-differentiable transformations of either the response variables $\mathbf{r}$ or the concentration variables $\mathbf{c}$ in our model leave the mutual information unchanged.

## Multiple glomeruli with the same affinity profile

In mammals, the axons from neurons expressing a given receptor type can project to anywhere from 2 to 16 different glomeruli. Here we show that in our setup, information transfer only depends on the total number of neurons of a given type, and not on the number of glomeruli to which they project.

The key observation is that mutual information, **Equation (3)** in the main text, is unchanged when the responses and/or concentrations are modified by invertible transformations (see previous section). In particular, linear transformations of the responses do not affect the information values. Suppose that we have a case in which two receptors $p$ and $q$ have identical affinities, so that $S_{pi} = S_{qi}$ for all odorants $i$. We can then form linear combinations of the corresponding glomerular responses,

$$
\begin{aligned}
r_+ &= r_p + r_q = (K_p + K_q) \sum_i S_{pi} c_i + \eta_p \sqrt{K_p} + \eta_q \sqrt{K_q}, \\
r_- &= K_q r_p - K_p r_q = \eta_p K_q \sqrt{K_p} - \eta_q K_p \sqrt{K_q},
\end{aligned}
\tag{56}
$$

and consider a transformation that replaces $(r_p, r_q)$ with $(r_+, r_-)$. Since $r_-$ is pure noise, that is it does not depend on the concentration vector $\mathbf{c}$ in any way, it has no effect on the mutual information.

We have thus shown that the amount of information that $M$ receptor types contain about the environment when two of the receptors have identical affinity profiles is the same as if there were only $M - 1$ receptor types. The two redundant receptors can be replaced by a single one with an abundance equal to the sum of the abundances of the two original receptors. The sum of two Gaussian variables with the same mean is Gaussian itself and has a variance equal to the sum of the variances of the two variables, meaning that the noise term $\eta_+$ in the $r_+$ response has variance $\frac{K_p \sigma_p^2 + K_q \sigma_q^2}{K_p + K_q}$.

## Appendix 3

DOI: https://doi.org/10.7554/eLife.39279.012

# A nonlinear response example

## Estimating the mutual information numerically

Consider an extension of our model in which the responses depend in a nonlinear way on concentrations, but are still subject to pure Gaussian noise:

$$\bar{r}_a = f_a(\mathbf{c}) + \frac{1}{\sqrt{K_a}}\eta_a, \qquad \eta_a \sim \mathcal{N}(0, \sigma_a^2). \tag{57}$$

Note that here we are calculating the average OSN response $\bar{r}_a = r_a/K_a$, while in the main text we used the total response $r_a$. As far as mutual information calculations are concerned, the difference between $\bar{r}_a$ and $r_a$ does not matter, as they are related by an invertible transformation.

Unless the functions $f_a$ are linear, a closed-form solution for the mutual information between concentrations and responses cannot be found. It is thus necessary to calculate the mutual information integral numerically. We can still do part of the calculation analytically, though:

$$
\begin{aligned}
I(\bar{\mathbf{r}}, \mathbf{c}) &= \int d^M\bar{r}\, d^N c\, P(\bar{\mathbf{r}}, \mathbf{c}) \log \frac{P(\bar{\mathbf{r}}|\mathbf{c})}{P(\bar{\mathbf{r}})} \\
&= -\int d^M\bar{r}\, P(\bar{\mathbf{r}}) \log P(\bar{\mathbf{r}}) + \int d^N c\, P(\mathbf{c})\, d^M\bar{r}\, P(\bar{\mathbf{r}}\mathbf{c}) \log P(\bar{\mathbf{r}}|\mathbf{c}).
\end{aligned}
\tag{58}
$$

In our case, $P(\bar{\mathbf{r}}|\mathbf{c})$ is a multivariate Gaussian distribution whose covariance matrix is $\Sigma\mathbb{K}^{-1}$ and does not depend on the concentrations. This means that the $\mathbf{c}$ integral in the second term can be performed independently of the $\bar{\mathbf{r}}$ integral, in which case it drops out of the calculation, as it is equal to 1. The $\bar{\mathbf{r}}$ integral is simply the negative entropy of a multivariate Gaussian distribution, and is thus equal to

$$
\begin{aligned}
\int d^M\bar{r}\, P(\bar{\mathbf{r}}|\mathbf{c}) \log P(\bar{\mathbf{r}}|\mathbf{c}) &= -\frac{1}{2}\log\det\Sigma\mathbb{K}^{-1} - \frac{M}{2}\log 2\pi e \\
&= -\frac{1}{2}\sum_a \log\left(2\pi e\, \frac{\sigma_a^2}{K_a}\right).
\end{aligned}
\tag{59}
$$

The first term in **Equation (58)** is the entropy of the responses, which needs to be calculated numerically. We use a histogram method, in which we split the space of possible responses along each dimension into bins of equal size $\Delta$. We then estimate the probability in each bin. If $i_1 \ldots i_M$ indexes the bins, we can then think of the response distribution as a discrete PDF $P_{i_1\ldots i_M}$, and we can estimate the entropy using

$$H(\bar{\mathbf{r}}) = -\int d^M\bar{r}\, P(\bar{\mathbf{r}}) \log P(\bar{\mathbf{r}}) \approx \sum_{i_1\ldots i_M} P_{i_1\ldots i_M} \log \frac{P_{i_1\ldots i_M}}{\Delta^M}. \tag{60}$$

In this approach, the challenge remains to estimate the PDF of the responses,

$$P(\bar{\mathbf{r}}) = \int d^N c\, P(\mathbf{c}) P(\bar{\mathbf{r}}|\mathbf{c}) = \frac{1}{\sqrt{(2\pi)^M \det\Sigma\mathbb{K}^{-1}}} \int d^N c\, P(\mathbf{c}) \exp\left[-\frac{1}{2}(\bar{\mathbf{r}} - \mathbf{f}(\mathbf{c}))^T \mathbb{K}\Sigma^{-1}(\bar{\mathbf{r}} - \mathbf{f}(\mathbf{c}))\right] \tag{61}$$

where $\mathbf{f}$ is the vector of response functions $\mathbf{f} = (f_1, \ldots, f_M)$. We do this using a sampling technique based on the law of large numbers. Given $n$ sample concentration vectors $c_i$ drawn from the probability distribution $P(\mathbf{c})$, we have

$$P(\bar{\mathbf{r}}) = \mathbb{E}_{P(\mathbf{c})}\{\frac{1}{\sqrt{(2\pi)^M \det \Sigma \mathbb{K}^{-1}}} \exp[-\frac{1}{2}(\bar{\mathbf{r}} - \mathbf{f}(\mathbf{c}))^T \mathbb{K}\Sigma^{-1}(\bar{\mathbf{r}} - \mathbf{f}(\mathbf{c}))]\}$$

$$\approx \frac{1}{n}\sum_i \frac{1}{\sqrt{(2\pi)^M \det \Sigma \mathbb{K}^{-1}}} \exp[-\frac{1}{2}(\bar{\mathbf{r}} - \mathbf{f}(\mathbf{c}_i))^T \mathbb{K}\Sigma^{-1}(\bar{\mathbf{r}} - \mathbf{f}(\mathbf{c}_i))], \tag{62}$$

where $\mathbb{E}_{P(\mathbf{c})}\{\cdots\}$ denotes the expected value under the distribution of concentrations. We use this formula to estimate the histogram elements $P_{i_1 \ldots i_M}$ and then use **Equation (60)** to estimate the response entropy $H(\bar{\mathbf{r}})$. We then plug $H(\bar{\mathbf{r}})$ and **Equation (59)** into **Equation (58)** to find the mutual information. Note that we have not assumed anything about the natural distribution of odor concentrations, $P(\mathbf{c})$, so that we are not restricted to Gaussian environments with this method.

## Competitive binding model

The way in which olfactory neurons respond to arbitrary mixtures of odorants is not completely understood. However, simple kinetic models in which different odorant molecules compete for the same receptor binding site have been shown to capture much of the observed behavior (**Singh et al., 2018**). In such models, the activation of an OSN of type $a$ in response to a set of odorants with concentrations $c_i$ is given by

$$r_a = \frac{\sum_i e_{ai}c_i/EC50_{ai}}{1 + \sum_i c_i/EC50_{ai}}, \tag{63}$$

where $EC50_{ai}$ is the concentration of odorant $i$ for which the response for the OSN of type $a$ reaches half its maximum, and $e_{ai}$ is the maximum response elicited by odorant $i$ in an OSN of type $a$.

## Results from a toy problem

The computation time from the method outlined above for calculating mutual information grows exponentially with the dimensionality $M$ of the response space. Additionally, it grows linearly with the number $n$ of samples drawn from the odor distribution, which in turn needs to grow exponentially with the number $N$ of odorants we are considering in order to sample concentration space sufficiently well. For this reason, large-scale simulations involving this method are infeasible.

Thus we focused on a simple example with $M = 3$ receptors and $N = 15$ odorants. We used an arbitrary subset of elements from the fly sensing matrix and a pair of randomly-generated non-overlapping environments (**Appendix 3—figure 1**) to first calculate the optimal receptor distribution using the linear method described in the main text (**Appendix 3—figure 2**, top). We chose the scale of the environment covariance matrices to get a variability in the responses of around 1, large enough to enter the nonlinear regime when using the nonlinear response function (described below). We then set the total neuron population to $K_{\mathrm{tot}} = 200$, which put us in an intermediate SNR regime in which all the receptor types were used in the optimal distribution, but their abundances were different (**Appendix 3—figure 2**, top).

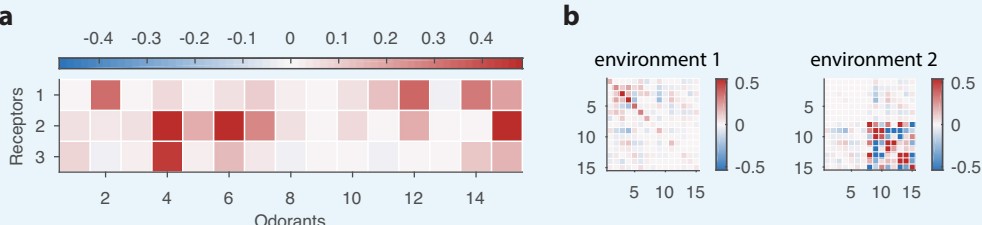

**Appendix 3—figure 1.** Sensing matrix and environment covariance matrices used in our toy

problem involving a non-linear response function.
DOI: https://doi.org/10.7554/eLife.39279.018

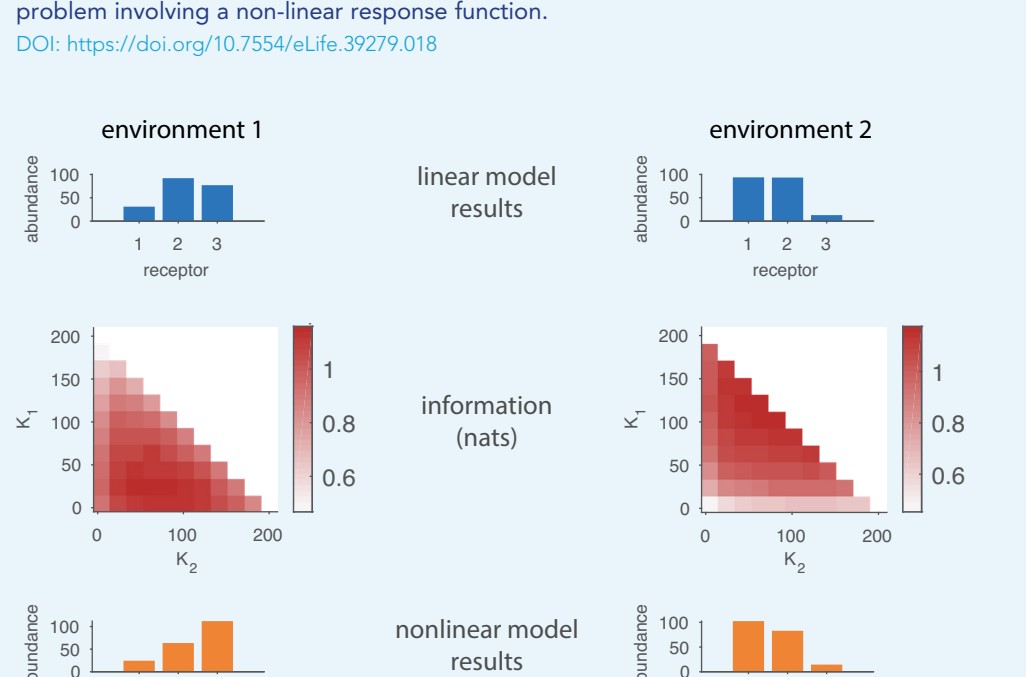

**Appendix 3—figure 2.** Comparing results from the linear model in the main text to results based on a nonlinear response function. The top row shows the optimal receptor distribution obtained using the linear model for a system with three receptor types and 15 odorants. The middle row shows how the estimated mutual information varies with OSN abundances in a nonlinear model based on a competitive binding response function. The bottom rows shows the optimal receptor distribution from the nonlinear model, obtained by finding the cells in the middle row in which the information is maximized.
DOI: https://doi.org/10.7554/eLife.39279.019

In the linear approximation, we found that receptor 1 is under-represented in environment 1, while in environment 2 receptor 3 has very low abundance. We wanted to see how much this result is affected by a nonlinear response function. We used a competitive binding model as described above in which the matrix of EC50 values was taken equal to the sensing matrix used in the linear case, and the efficacies $e_{ai}$ were all set to 1:

$$r_a = \frac{\sum_i S_{ai} c_i}{1 + \sum_i S_{ai} c_i} + \frac{1}{\sqrt{K_a}} \eta_a .$$

(64)

To calculate the mutual information between responses and concentrations for a fixed choice of neuron abundances $K_a$, we used the procedure outlined above with 20 bins between –0.75 and 1.5 for each of the response dimensions. We sampled $n = 10^4$ concentration vectors to build the response histogram. We calculated the information values in both environments at a $10 \times 10$ grid of OSN abundances (**Appendix 3—figure 2**, middle row), and found the cell which maximized the information. The OSN abundances at this maximum (**Appendix 3—figure 2**, bottom) show the same pattern of change as we found in the linear approximation, with receptors 1 and 3 exchanging places as least abundant in the OSN population.

## Appendix 4

DOI: https://doi.org/10.7554/eLife.39279.012

# Random environment matrices

## Generating random covariance matrices

Generating plausible olfactory environments is difficult because so little is known about natural odor scenes. However, it is reasonable to expect that there will be some strong correlations. This could, for instance, be due to the fact that an animal's odor is composed of several different odorants in fixed proportions, and thus the concentrations with which these odorants are encountered will be correlated.

The most straightforward way to generate a random covariance matrix would be to take the product of a random matrix with its transpose, $\Gamma = MM^T$. This automatically ensures that the result is positive (semi)definite. The downside of this method is that the resulting correlation matrices tend to cluster close to the identity (assuming that the entries of $M$ are chosen *i.i.d.*). One way to avoid this would be to use matrices $M$ that have fewer columns than rows, which indeed leads to non-trivial correlations in $\Gamma$. However, this only generates rank-deficient covariance matrices which means that odorant concentrations are constrained to live on a lower-dimensional hyperplane. This is too strong a constraint from a biological standpoint.

To avoid these shortcomings, we used a different approach for generating random covariance matrices. We split the process into two parts: we first generated a random correlation matrix by the method described below, in which all the variances (i.e. the diagonal elements) were equal to 1; next we multiplied each row and corresponding column by a standard deviation drawn from a lognormal distribution.

In order to generate random correlation matrices, we used a modified form of an algorithm based on partial correlations (*Lewandowski et al., 2009*). The partial correlation between two variables $X_i$ and $X_j$ conditioned on a set of variables $L$ is the correlation coefficient between the residuals $R_i$ and $R_j$ obtained by subtracting the best linear fit for $X_i$ and $X_j$ using all the variables in $L$. In other words, the partial correlation between $X_i$ and $X_j$ is equal to that part of the correlation coefficient that is not explained by the two variables depending on a common set of explanatory variables, $L$. In our case the $X_i$ are the concentrations of different odorants in the environment and the partial correlations in question are, for example, the correlation between any pair of the odorants conditioned on the remaining ones. We want to construct the unconditioned correlation matrix between the odor concentrations vectors of the environment. There is an algorithm to construct this matrix that starts by randomly drawing the partial correlation between the first two odorants $X_1$ and $X_2$ conditioned on the rest, and then recursively reducing the size of the conditioning set while generating more random partial correlations until the un-conditioned correlation values are obtained. For details, see *Lewandowski et al. (2009)*.

The specific procedure used in *Lewandowski et al. (2009)* draws the partial correlation values from beta distributions with parameters depending on the number of elements in the conditioning set $L$. This is done in order to ensure a uniform sampling of correlation matrices. This, however, is not ideal for our purposes because these samples again tend to cluster close to the identity matrix. A simple modification of the algorithm that provides a tunable amount of correlations is to keep the order of the beta distribution fixed $\alpha = \beta = \mathrm{const}$ (see Stack Exchange, at https://stats.stackexchange.com/q/125020). When the parameter $\beta$ is large we obtain environments with little correlation structure, while small $\beta$ values lead to stronger correlations between odorant concentrations. The functions implementing the generation of random environments are available on our GitHub (RRID:SCR_002630) repository at https://github.com/ttesileanu/OlfactoryReceptorDistribution (see environment/generate_random_environment.m and utils/randcorr.m).

## Perturbing covariance matrices

When comparing the qualitative results from our model against experiments in which the odor environment changes (*Ibarra-Soria et al., 2017*), we used small perturbations of the initial and final environments to estimate error bars on receptor abundances. To generate a perturbed covariance matrix, $\tilde{\Gamma}$, from a starting matrix $\Gamma$, we first took the matrix square root: a symmetric matrix $M$, which obeys

$$\Gamma = MM^T \equiv M^2 . \tag{65}$$

We then perturbed $M$ by adding normally-distributed *i.i.d.* values to its elements,

$$\tilde{M}_{ij} = M_{ij} + \sigma \eta_{ij} , \tag{66}$$

and recreated a covariance matrix by multiplying the perturbed square root with its transpose,

$$\tilde{\Gamma} = \tilde{M}\tilde{M}^T . \tag{67}$$

This approach ensures that the perturbed matrix $\tilde{\Gamma}$ remains a valid covariance matrix—symmetric and positive-definite—which would not be guaranteed if the random perturbation was added directly to $\Gamma$. We chose the magnitude $\sigma$ of the perturbation so that the error bars in our simulations are of comparable magnitude to those in the experiments.

We used a similar method for generating the results from **Figure 3**, where we needed to apply the same perturbation to two different environments. Given the environment covariance matrices $\Gamma_k$, with $k \in \{1, 2\}$, we took the matrix square root of each environment matrix, $M_k = \Gamma_k^{1/2}$. We then added the same perturbation to both, $\tilde{M}_k = M_k + P$, then recovered covariance matrices for the perturbed environments by squaring $\tilde{M}_k$, $\tilde{\Gamma}_k = \tilde{M}_k\tilde{M}_k^T$. In the examples used in the main text, the perturbation $P$ was a matrix in which only one column was non-zero. The elements in this column were chosen from a Gaussian distribution with zero mean and a standard deviation five times larger than the square root of the median element of $\Gamma_1$. This choice was arbitrary and was made to obtain a visible change in the optimal receptor abundances between the 'control' and 'exposed' environments.

Finally, we employed this approach also for generating non-overlapping environments. Given two environments $\Gamma_1$ and $\Gamma_2$ and their matrix square roots $M_1$ and $M_2$, we reduced the amount of variance in the first half of $M_1$'s columns and in the second half of $M_2$'s. We did this by dividing those columns by a constant factor $f$, which in this case we chose to be $f = 4$. We then used the resulting matrices $\tilde{M}_k$ to generate covariance matrices $\tilde{\Gamma}_k = \tilde{M}_k\tilde{M}_k^T$ with largely non-overlapping odors.

## Appendix 5

DOI: https://doi.org/10.7554/eLife.39279.012

### Deriving the dynamical model

To turn the maximization requirement into a dynamical model, we employ a gradient ascent argument. Given the current abundances $K_a$, we demand that they change in proportion to the corresponding components of the information gradient, plus a Lagrange multiplier to impose the constraint on the total number of neurons:

$$\dot{K}_a = 2\alpha\left(\frac{\partial I}{\partial K_a} - \lambda\right) = \alpha\left[(\tilde{Q}^{-1} + \mathbb{K})^{-1}_{aa} - \lambda\right]. \tag{68}$$

The brain does not have direct access to the overlap matrix $Q$, but it could measure the response covariance matrix $R$ from *Equation (13)*. Thus, we can write the dynamics as

$$\begin{aligned}
\dot{K}_a &= \alpha\{[\tilde{Q}(\mathbb{I} + \mathbb{K}\tilde{Q})^{-1}]_{aa} - \lambda\} \\
&= \alpha\{[\mathbb{K}^{-1}(\Sigma^{-1/2}R\mathbb{K}^{-1}\Sigma^{-1/2} - \mathbb{I})\Sigma^{1/2}\mathbb{K}R^{-1}\Sigma^{1/2}]_{aa} - \lambda\} \\
&= \alpha\{K_a^{-1} - \lambda - (\Sigma^{1/2}R^{-1}\Sigma^{1/2})_{aa}\} \\
&= \alpha\{K_a^{-1} - \lambda - \sigma_a^2 R_{aa}^{-1}\},
\end{aligned} \tag{69}$$

where we used the fact that $\Sigma^{1/2}$ and $\mathbb{K}$ are diagonal and thus commute. These equations do not yet obey the non-negativity constraint on the receptor abundances. The divergence in the $K_a^{-1}$ term would superficially appear to ensure that positive abundances stay positive, but there is a hidden quadratic divergence in the response covariance term, $R_{aa}^{-1}$; see *Equation (13)*. To ensure that all constraints are satisfied while avoiding divergences, we multiply the right-hand-side of *Equation (69)* by $K_a^2$, yielding

$$\dot{K}_a = \alpha[K_a - K_a^2(\lambda + \sigma_a^2 R_{aa}^{-1})], \tag{70}$$

which is the same as *Equation (9)* from the main text.

If we keep the Lagrange multiplier $\lambda$ constant, the asymptotic value for the total number of neurons $K_{\text{tot}}$ will depend on the statistical structure of olfactory scenes. If instead we want to enforce the constraint $\sum K_a = K_{\text{tot}}$ for a predetermined $K_{\text{tot}}$, we can promote $\lambda$ itself to a dynamical variable,

$$\frac{d\lambda}{dt} = \beta\left[\sum_a K_a - K_{\text{tot}}\right], \tag{71}$$

where $\beta$ is another learning rate. Provided that the dynamics of $\lambda$ is sufficiently slow compared to that of the neuronal populations $K_a$, this will tune the experience-independent component of the neuronal death rate until the total population stabilizes at $K_{\text{tot}}$.

## Appendix 6

DOI: https://doi.org/10.7554/eLife.39279.012

### Interpretation of diagonal elements of the inverse overlap matrix

In the main text we saw that the diagonal elements of the inverse overlap matrix $Q_{aa}^{-1}$ were related to the abundances of OSNs $K_a$. Specifically,

$$K_a \approx \frac{1}{\lambda} - \sigma_a^2 Q_{aa}^{-1}, \tag{72}$$

where $\lambda$ is a Lagrange multiplier imposing the constraint on the total number of neurons. As noted around **Equation (13)** above, the overlap matrix $Q$ is related to the response covariance matrix $R$: in particular, $Q$ is equal to $R$ when there is a single receptor of each type ($K_a = 1$) and there is no noise ($\sigma_a = 0$). That is, the overlap matrix measures the covariances between responses in the absence of noise. This means that its inverse $A = Q^{-1}$ is effectively a so-called 'precision matrix'. Diagonal elements of a precision matrix are inversely related to corresponding diagonal elements of the covariance matrix (i.e. the variances), but, as we will see below, they are also monotonically related to parameters that measure how well each receptor response can be linearly predicted from all the others. Since receptor responses that either do not fluctuate much or whose values can be guessed based on the responses of other receptors are not very informative, we would expect that abundances $K_a$ are low when the corresponding diagonal elements of the inverse overlap matrix $A_{aa}$ are high, which is what we see. In the following we give a short derivation of the connection between the diagonal elements of precision matrices and linear prediction of receptor responses.

Let us work in the particular case in which there is one copy of each receptor and where there is no noise, so that $Q = R$, that is $Q_{ij} = \langle r_i r_j \rangle - \langle r_i \rangle \langle r_j \rangle$. Without loss of generality, we focus on calculating the first diagonal element of the inverse overlap matrix, $A_{11}$, where $A = Q^{-1}$. For notational convenience, we will also denote the mean-centered first response variable by $y \equiv r_1 - \langle r_1 \rangle$, and the subsequent ones by $x_a \equiv r_{a+1} - \langle r_{a+1} \rangle$. Then the covariance matrix $Q$ can be written in block form

$$Q = \begin{pmatrix} \langle y^2 \rangle & \langle y\mathbf{x}^T \rangle \\ \langle y\mathbf{x} \rangle & M \end{pmatrix}, \tag{73}$$

where $M$ is

$$M = \langle \mathbf{x}\mathbf{x}^T \rangle, \tag{74}$$

and $\mathbf{x}$ is a column vector containing the $x_a$ variables. Using the definition of the inverse together with Laplace's formula for determinants, we get

$$A_{11} = \frac{\det M}{\det Q}. \tag{75}$$

Using the Schur determinant identity (derived above) on the block form (**Equation (73)**) of the matrix $Q$,

$$\begin{aligned} A_{11} &= \frac{\det M}{\det M \cdot \det[\langle y^2 \rangle - \langle y\mathbf{x}^T \rangle M^{-1} \langle y\mathbf{x} \rangle]} \\ &= \frac{1}{\langle y^2 \rangle - \langle y\mathbf{x}^T \rangle M^{-1} \langle y\mathbf{x} \rangle}, \end{aligned} \tag{76}$$

where we used the fact that the argument of the second determinant is a scalar.

Now, consider approximating the first response variable $y$ by a linear function of all the others:

$$y = \mathbf{a}^T \mathbf{x} + q, \tag{77}$$

where $q$ is the residual. Note that we do not need an intercept term because we mean-centered our variables, $\langle y \rangle = \langle x \rangle = 0$. Finding the coefficients a that lead to a best fit (in the least-squares sense) requires minimizing the variance of the residual, and a short calculation yields

$$\mathbf{a}^* = \arg\min_{\mathbf{a}} \langle q \rangle^2 = \arg\min_{\mathbf{a}} (y - \mathbf{a}^T \mathbf{x})^2 = M^{-1} \langle y\mathbf{x} \rangle, \tag{78}$$

where $M$ is the same as the matrix defined in **Equation (74)**.

The coefficient of determination $\rho^2$ is defined as the ratio of explained variance to total variance of the variable $y$,

$$\begin{aligned} \rho^2 \quad &= \frac{\langle (\mathbf{a}^{*T} \mathbf{x})^2 \rangle}{\langle y^2 \rangle} = \frac{\mathbf{a}^{*T} \langle \mathbf{xx}^T \rangle \mathbf{a}^*}{\langle y^2 \rangle} = \frac{\langle y\mathbf{x}^T \rangle M^{-1} M M^{-1} \langle y\mathbf{x} \rangle}{\langle y^2 \rangle} \\ &= \frac{\langle y\mathbf{x}^T \rangle M^{-1} \langle y\mathbf{x} \rangle}{\langle y^2 \rangle}. \end{aligned} \tag{79}$$

Comparing this to **Equation (76)**, we see that

$$A_{11} = \frac{1}{\langle y^2 \rangle} \frac{1}{1 - \rho^2}, \tag{80}$$

showing that the diagonal elements of the precision matrix are monotonically related to the goodness-of-fit parameter $\rho^2$ that indicates how well the corresponding variable can be linearly predicted by all the other variables. In addition, the inverse dependence on the variance of the response $\langle y \rangle^2$ shows that variables that do not fluctuate much (low $\langle y \rangle^2$) lead to high diagonal values of the precision matrix . From **Equation (72)**, we see that these variances should be considered 'large' or "small" in comparison with the noise level in each receptor ($\sigma_a$). Since receptor responses that either do not fluctuate much or whose values can be guessed based on the responses of other receptors are not very informative, we should find that receptor abundances $K_a$ are low when the corresponding diagonal elements of the inverse overlap matrix $A_{aa} = Q_{aa}^{-1}$ are high.

