## [Decision Letter]

Thank you for submitting your article "Adaptation of olfactory receptor abundances for efficient coding" for consideration by *eLife*. Your article has been reviewed by 3 peer reviewers, one of whom is a member of our Board of Reviewing Editors, and the evaluation has been overseen by a Senior Editor. The following individual involved in review of your submission has agreed to reveal his identity: David Zwicker (Reviewer #3).

The reviewers have discussed the reviews with one another and the Reviewing Editor has drafted this decision to help you prepare a revised submission.

Summary:

In this manuscript Tesileanu and colleagues present a theoretical analysis of optimal coding in olfactory systems. They derive analytical results and use simulations to ask how receptor distributions depend on the number of neurons, the tuning width of receptors, and environment, with a core assumption of efficient coding. The study leads to the interesting prediction of strikingly changed receptor distribution following olfactory experience.

Essential revisions:

The current paper lays out a good framework but would be much stronger if some essential ramifications of the core idea were to be addressed.

1) The model must make predictions that can be falsified by experimental or evolutionary data.

2) The authors should incorporate more biological activation functions and receptor sensitivity distributions and examine how these affect the conclusions of the model.

3) The authors should comment on the diversity of olfactory systems across evolution and note how their model does or does not account for this diversity.

4) The authors should address the question of what happens when the number of receptors changes (as opposed to the number of neurons), as this is one of the main variables that seems to differ across evolution.

*Reviewer #1:*

This study builds its analysis on the idea that olfactory coding lies in a regime where sensor responses are correlated, hence efficient coding leads to divergent receptor abundance.

In addition to the assumption of efficient coding, the manuscript also assumes that olfactory receptor populations adapt to achieve such coding, within the time-frame of receptor turnover. This assumption leads to the interesting prediction of strikingly changed receptor distribution following olfactory experience, a phenomenon that has been observed experimentally.

I find the analysis interesting and potentially insightful, but it misses out on a few key biological points, that I feel really should be taken on board if the analysis is to be biologically relevant. I'll enumerate three of these, in increasing order of concern.

1) The authors explicitly ignore temporal correlations in olfactory cues, with a brief line in the introduction to their model that states that spike timing could be incorporated into the model. I do not see how this will work for respiratory phase tuning of odor responses, and would be interested to see what the authors had in mind for this.

2) The authors choose an operating point where they can apply a linear model for glomerular responses. In the animal, the operating range of different receptors for different odors is rather diverse, with the half-max varying substantially and the slope also varies. Thus a subset of odors will be saturating for some receptors, but linear or even subthreshold for other receptors. I suspect that this will affect the analysis of the responses.

My view is that any coding theory has to account for the very wide range of odor concentrations encountered in nature. One could possibly add this to the analysis reported in Equations 4 to 6, by summing the mutual information over a set of odor ranges, in which different but overlapping subsets of receptors are involved. I would be interested to see if this alters the conclusions.

3) A major point of concern with the whole analysis is of salience. The obvious outlier here is pheromones. Enormous resources are allocated to pheromone detection, and clearly this doesn't seem to fall within the framework presented in the paper. Even with the general olfactory system, the assumption of efficient coding needs to be further mapped to the distribution of odor salience, that is, relevance for animal survival. There seems to be a subtle nod to this point in the third-last paragraph of the Discussion, where 'value of detecting different odorants' is mentioned. I feel that the point is central enough that it needs to be fully addressed.

The constraint is not just to efficiently code the environment, it is to efficiently code those aspects of the environment which matter for survival. This seems to give rise to a fundamental challenge to this model, as follows: Assume a rare predator with a characteristic odor. Even if the predator is absent from the odor scene for long periods, it would be fatal to the prey species to underexpress receptors sensitive to the predator. One can come up with numerous other examples on these lines where selection pressures necessitate receptor expression for reasons other than efficient coding. There may be a couple of ways to go about incorporating this into the model: an evolutionarily determined 'prior' that weights salience of receptors, or a more general rule that tries to ensure a certain degree of broad coverage even at the expense of efficient coding. I suspect both may be relevant.

In summary, I think that the current paper lays out a good framework but would be much stronger if some essential ramifications of the core idea were to be addressed.

*Reviewer #2:*

In this manuscript Tesileanu and colleagues present a theoretical analysis of optimal coding in olfactory systems. The goal is to find the distribution of olfactory receptor abundances that maximizes the information an olfactory system can gain about odors in its environment, and to predict how receptor abundances should change when the environment changes. Given a set of assumptions about how odors are encoded by a population of receptors, they derive an expression for the mutual information between the response of a receptor population and a vector of environmental odors. They then evaluate this expression and show that the information depends on an overlap matrix, related to the covariance of the environmental odor vector. Based on these analytical results, they use simulations to ask how receptor distributions depend on the number of neurons, and the tuning width of receptors. They then ask how receptor abundances should change when the environment changes. They report a number of findings: (1) Receptor abundances are more sensitive to environmental perturbations when the number of neurons is small or intermediate, (2) Receptor abundances are more sensitive to environmental perturbations when they are narrowly tuned, (3) changes in optimal receptor abundances cannot be simply predicted from changes in odor abundances or variances.

At an abstract level, olfactory systems can be thought of as arrays of receptors, which have evolved from distinct receptor families many times over the course of evolution. Olfactory receptor genes are among the largest and most rapidly evolving gene families. Therefore I highly support the goals of this study to provide a theoretical understanding of how receptor arrays should change in response to changes in odor environment. In general, the level of abstraction adopted in this study is appropriate, and some of the findings are interesting. However, I have a number of questions about the analyses performed and conclusions reached, particularly concerning how the results might be related to biologically testable phenomena.

1) The conclusions concerning how receptor abundances should change following a change in environment are disappointing. While their model recapitulates Ibarra-Soira's result which predicts that the distribution of high abundance receptors is likely to remain unchanged, they do not provide any concrete predictions on the receptors which change their abundance in either direction of change or magnitude. As currently stated, the central predictions of the model – that optimal receptor abundances can increase or decrease or stay the same following a change in environment – seems to be unfalsifiable.

The manuscript could be strengthened by making more concrete predictions about how receptor abundances should change, at least in particular regimes. For example, the authors note that for intermediate numbers of neurons, optimal receptor distributions are anti-correlated with the inverse of the overlap matrix Q^-1^. They expand on this to say that receptors with high Q^-1^ can be uninformative because they do not fluctuate or because they provide redundant information. Although I did not fully follow the arguments here, it seemed like this was saying that abundance is inversely related to information, and there are two ways to be uninformative, one by having low variance, and two by being highly correlated with other receptors. Could this be used to make more concrete predictions about predicted changes in receptor abundance, at least for a given number of neurons? In addition, the authors also provide model evidence for predicting the magnitude of the change based on the change in olfactory environment, but it is unclear the characteristics which group types of changes together.

2) Some of the conclusions seem odd when considered in the context of olfactory evolution. For example, the authors conclude that if the number of neurons is large, then the optimal receptor distribution is approximately uniform. Olfactory systems differ greatly in magnitude across organisms. In particular, two of the most-studied models, fly and mouse, differ by an order of magnitude in the number of receptors (~60 for fly, ~1000 for mouse), as well as the total number of neurons. The finding that total neuron number determines receptor distribution should be tied numerically to the olfactory systems of flies and mice, if not also for other organisms. It is unclear, for example, whether the olfactory receptor number of mice is considered large, or whether it would fall in the intermediate signal to noise regime. Does the model predict that mouse receptor distributions are uniform while fly distributions are highly skewed? Why then is any adaptation observed in mouse receptor abundances as has been observed experimentally?

Given the results presented here one might imagine that the optimal strategy would be to make a very large number of broadly tuned receptors. Instead, what we observe across evolution are olfactory systems of various sizes, with various widths of odor tuning, all constantly evolving. The number of receptors in particular seems to be under strong evolutionary pressure, with new gene families expanding (as in ant ORs) or collapsing (as in humans). This discrepancy, or the other constraints that might lead to the biological situation, should be commented on.

The authors state that receptor abundances do not change in insects and therefore focus on a mammalian example to test their hypothesis. However, insect olfactory systems evolve quite rapidly between closely related species, and there is a large literature on this, especially from the Hansson group (e.g. Dekker…Hansson, 2006). Can these studies be used to test any of the hypotheses here? Or can the authors propose comparative studies that would test their hypotheses?

3) Several concepts used in the text are a bit unclear, at least to a biological reader:

Could the authors provide some intuition for what is meant (biologically) by the inverse of the overlap matrix?

Could the authors please unpack the following sentence:

The quantity KQ thus behaves as a signal-to-noise ratio (SNR), so that Equation 4 is essentially a generalization to multiple, correlated channels of the standard result for a single Gaussian channel, I = 1 log(1 + SNR^2^).

Could the authors please clarify in the discussion of Equation 7 whether *K_tot_* represents the total number of neurons, the number of receptors, or the number of receptor types? Is the total number of neurons the most sensible thing to vary or would it be interesting to look at olfactory systems with different numbers of receptor types? This seems related to the question of where noise arises in the system, and what other constraints, besides information as quantified here, an animal might have on the design of its olfactory system.

4) The investigation of how optimal coding changes with broad versus narrow tuned receptors was interesting. However, real receptor arrays, at least as seen in the Hallem data, contain a mix of broadly and narrowly-tuned receptors, and receptor tuning width depends on odor intensity, with many receptor showing narrowly tuned response at low concentrations and wider tuning at high concentrations. Could the authors explore what happens in this regime, and provide any explanation for why animals might have both broad and narrowly tuned receptors? This finding could be further explored by making predictions for olfactory systems with receptors of mixed tuning widths, as is generally accepted to be the case in most organisms. This would provide a more concrete prediction for future experiments.

5) The authors claim that their model is robust to non-linearities and as well as their choice to represent the olfactory environment as a vector of concentrations. These ideas should be tested and demonstrated within the paper. For example, the nonlinearities involved in receptor encoding are well known: receptor responses can be expressed as a Hill function of odor concentration:

r = (c^n)/(c^n+Kd)

In many olfactory systems n=1, further simplifying this equation. The authors should explicitly show that the model generalizes when this nonlinearity is included. In addition, the main sources of noise in receptor encoding are likely to be (1) difference in receptor abundance across neurons that express the same receptor, (2) stochasticity in receptor binding and activation. The authors might consider incorporating these sources noise and showing that the model extends in this case.

The first section of the Results is difficult to read because it contained a number of statements justifying elements of the model and claiming that these do not affect the conclusions. This section would be easier to read if these points were saved for later in the manuscript where they could be explicitly demonstrated.

6) The section on dynamical optimization at the end seemed least well-constrained by data, and also (as noted) somewhat preliminary. The authors might consider reserving this material for a future manuscript that explores dynamics and tests them more thoroughly, and instead using this space to show that the model still holds when certain assumptions in the first version of the model are relaxed.

7) The authors should consider including graphical representations, similar to those provided in Figure 1, for concepts such as the mutual information measure, the covariance matrix, the overlap matrix, and the inverse overlap matrix. This would help provide insight for readers with less mathematical background, who may nonetheless be interested in the predictions of the models.

*Reviewer #3:*

The paper investigates theoretically how changing copy numbers of olfactory sensory neurons affects the coding properties of the olfactory system. The authors introduce a simple model based on the maximization of mutual information, which they analyze analytically and numerically using both artificial and measured values for the receptor sensitivities. Their analysis reveals a complex dependence of the optimal copy numbers of expressed receptors on the correlation structure of the receptor sensitivities and the odor environment. Since qualitatively similar dependencies have been observed in experiments, the model is very valuable for understanding the dynamics of copy number adaptation in the olfactory system. More generally, the presented model of the olfactory system is helpful for discussing how sensory systems adapt to changes in the environment and whether the aim for efficient coding is the driving mechanism.

The manuscript is well written and the arguments are clearly presented for the most part. My main concerns with the manuscript are that some limitations are not spelled out explicitly and that the theoretical analysis could have been more comprehensive. In particular, the authors do not investigate how their model would fair in the realistic case where odors are sparse and they do not discuss how the results depend on the number of different receptor types and the number of different odor molecules. The latter might be important to assess how relevant the results would be for realistic situations, since the current analysis is necessarily restricted to smaller numbers for the lack of adequate experimental data.

Taken together, I believe that the manuscript provides a substantial advance of our understanding of the olfactory system and of the adaptation of sensory systems to changing environments in general. I can therefore recommend publication of the manuscript in *eLife* once my comments have been taken into account.

---

## [Author Response]

Essential revisions:The current paper lays out a good framework but would be much stronger if some essential ramifications of the core idea were to be addressed.1) The model must make predictions that can be falsified by experimental or evolutionary data.

Our model makes both qualitative and quantitative predictions, which we have now highlighted in the paper, along with strategies for testing them.

First, our model makes qualitative predictions. For example, the number of receptor types should grow with the number of neurons in the olfactory epithelium, all else being equal, at least for closely-related species in similar ecological niches. Testing this prediction requires surveys of the number of receptor types and OSNs in different species, data which is currently fragmentary. As a step towards a test we plotted the number of intact OR genes in several mammalian species against an estimate of the number of OSNs derived from measures of the area of the olfactory epithelium and an allometric scaling law relating neural density and body mass (new Figure 2F). The trend is consistent with our predictions; a precise match is not expected since the species for which we found data live in different ecological niches and have presumably evolved distinct receptor repertoires.

Second, our model makes fully quantitative predictions for the abundances of olfactory neurons of different types, given receptor affinities and the statistics of the odor environment. Likewise, the model makes detailed predictions for how the abundances should change when the olfactory environment is modified. The predictions can be experimentally checked, for example using a protocol like that from Ibarra-Soria et al., 2017. We now describe this in detail in the section “A framework for a quantitative test”, which describes a procedure for working out the predictions in a given setting. This section is paired with a Matlab script that allows an interested researcher to plug measured affinity data and environmental statistics into our model and obtain numeric predictions for OSN abundances. We also applied our procedure to an in-silicoexperiment imitating Ibarra-Soria et al., 2017 with an available panel of 59 mouse and human receptors responding to 63 odorants. The results (Figure 6) qualitatively resemble the outcome of in vivoexperiments in mouse (~1000 receptor types responding to a complete olfactory environment). Finally, we investigated the robustness of our model’s predictions to sub-sampling of the receptors and the odor environment (new Figure 7, and accompanying discussion). We find that the predictions of absolute receptor abundances in an environment are robust to sub-sampling. Of course, more complete measurements will be required for predictions of smaller differences in receptor abundances between different environments.

Olfaction is a complex sense with many receptors sensing diverse odorants. Because of this, olfactory neuroscience lags behind visual neuroscience in the characterization of complete receptor repertoires and of natural olfactory scenes. However, large scale surveys of such data have begun, sponsored partly by the BRAIN program in the USA and by the NSF Olfaction Ideas Lab. Techniques are certainly available – e.g., mass spectrometry of volatile molecules harvested in a given environment. Our theoretical work motivates such large-scale surveys, and, given the data, will make precise predictions for new experiments.

Finally, in addition to these avenues for new experimental tests, our work is, to our knowledge, the first to propose a normative explanation for the observed qualitative behavior of receptor abundances in the olfactory epithelium including: (1) the inhomogeneous receptor distribution in the OSN population, and (2) the reproducible but apparently sporadic patterns of adaptation in receptor abundances following olfactory experience in mammals.

2) The authors should incorporate more biological activation functions and receptor sensitivity distributions and examine how these affect the conclusions of the model.

A challenge here is that the experimental data on receptor sensitivity distributions and biological activation functions is limited. To answer this question, we have leveraged available datasets surveying receptor responses to panels of odorants, as well as existing studies of response nonlinearities.

We are using receptor sensitivity values from fly (*Drosophila*) and from mammals (mouse and human) in the figures of the main text in addition to artificial sensitivity distributions with scalable tuning widths. In the updated Appendix (Appendix 1, Figures 1, 2, 3) we also include scrambled versions of these sensitivity distributions and additional artificial sensing matrices to show that the qualitative conclusions are robust to the details.

Fully including non-linear effects in OSN responses requires data from new experiments. Indeed, dose-response curves for neurons responding to single odorants are only available in a small number of cases, and nonlinearities in mixture responses that we would need in general are only beginning to be understood. There is some evidence (e.g., Singh et al., 2018, now cited in the paper) that a simple competitive binding model might give a reasonably good description of mixture responses in many cases. Following the reviewer’s suggestion, we used this framework as the starting point for the nonlinear results that we added to the Appendix (Appendix 3, A nonlinear response example). Such a model needs data on Hill coefficients and maximum activation values for every pair of receptor and odorant, and we estimated these from data in the fly. In the nonlinear case the mutual information must be numerically estimated, and, doing this, we found that in a simple example the qualitative structure of the results was the same as in a linear sensing model based on the same receptor data.

It is worth adding that typical neural nonlinearities show an approximately linear regime between the activation threshold and saturation. Our model should be regarded as a linearized approximation of this regime. Also, the mutual information that we optimize is invariant under invertible, smooth nonlinearities (Appendix 2, section “Invariance of mutual information under invertible and differentiable transformations”). For these reasons we expect our linear sensing model to provide a reasonable approximation which can be numerically extended to a fully nonlinear model when such data become broadly available for OSNs.

At a technical level, calculating the mutual information outside the linear and Gaussian idealization that we used is much more difficult because the required integrals must be calculated numerically, as we now describe in Appendix 3. The runtime for the simple code that we used in this case is orders of magnitude slower than that for the linear case and, worse, it grows exponentially with the number of receptor types and the number of odorants used in the problem. There are more advanced methods for estimating mutual information numerically, and there may be new approximation schemes that are better suited for our problem, but these are entire research projects in their own right and are beyond the scope of the present work. A related point is that entropy estimation and maximization are inherently difficult computationally, and so neural circuits might have no choice but to only approximately adapt to natural statistics.

On a broader methodological level, we feel that an important role of theory is to try to find aspects of biological systems that are “universal”, in the sense that the behavior of the system is roughly the same independent of microscopic details. We implicitly make use of such universality when we study olfaction without explicitly modeling the interaction between every molecule in the nasal epithelium and every volatile molecule that reaches it. Our premise is that capturing just the rough aspects of receptor responses as we do in our model might be enough to get a first approximation of the receptor abundances. The theoretical model can then be improved upon comparing the results from this simplified analysis to experiment.

3) The authors should comment on the diversity of olfactory systems across evolution and note how their model does or does not account for this diversity.

Our study focuses on the question of how to optimally use an available repertoire of olfactory receptors. We therefore take the set of available receptors, as well as their affinities to odorants, to be fixed. We do not seek to say anything about how these evolve.

Our model does suggest that larger olfactory systems (more OSNs in the epithelium), should support a greater diversity of receptor types. Strictly speaking the prediction is that, given a fixed repertoire of receptor types and olfactory environment, the number of types that are expressed should increase with the number of OSNs. Of course, even related animal species can typically have different genetically encoded receptor types and occupy different environmental niches. Unfortunately, information about the receptor repertoires and olfactory environments of different species is fragmentary and sometimes non-existent. Nevertheless, our theory leads to expect a general trend of receptor diversity increasing with OSN numbers. As a preliminary study, we illustrate this trend for some mammalian species in the new Figure 2F.

Receptor abundances can change faster than affinity profiles – in mammals, this even happens during the lifetime of an individual. Thus, by focusing on understanding the receptor abundances, we in effect focus on questions of adaptation on shorter timescales. There are other recent studies that approach the question of the evolution of receptor genes (e.g., Zwicker et al., 2016), but we are considering a different, complementary question here.

We edited the text to make these points clearer.

4) The authors should address the question of what happens when the number of receptors changes (as opposed to the number of neurons), as this is one of the main variables that seems to differ across evolution.

It is indeed interesting to see how our results change when we change the number of receptors (and also the number of odorants). We have now added results that show how the optimal abundances of the remaining receptors change when a fraction of the receptors is removed (new Figure 7).

The same results can also be interpreted in terms of the robustness of our results to incomplete sampling of the receptors. We find that even without having measurements for the affinity profiles for all receptor types, we can still get reasonable estimates for the optimal abundances of the receptors we do have data for. We also showed that a similar robustness holds for subsampling of odorants. This suggests that we can obtain reasonable results even without recording the affinity profile against every odorant in an environment, which would be difficult to achieve.

That said, we wish to emphasize again that we are not trying to build an evolutionary model of the olfactory periphery. We are mainly interested in changes that occur either during the lifetime of an individual, or on short evolutionary time periods, during which it may be easier to alter the abundances of receptors rather than their affinities. It thus seems reasonable to assume that the receptor types are fixed while optimizing the mutual information. Despite this, as we show in Figure 2, our model predicts that in certain regimes the receptor distribution will be inhomogeneous, so that some receptor types will be used in very small numbers. We would predict that the corresponding receptor genes will be more likely to undergo loss-of-function mutations.

Reviewer #1:[…] I find the analysis interesting and potentially insightful, but it misses out on a few key biological points, that I feel really should be taken on board if the analysis is to be biologically relevant. I'll enumerate three of these, in increasing order of concern.1) The authors explicitly ignore temporal correlations in olfactory cues, with a brief line in the introduction to their model that states that spike timing could be incorporated into the model. I do not see how this will work for respiratory phase tuning of odor responses, and would be interested to see what the authors had in mind for this.

One way in which temporal correlations could be implemented in our model would be to consider the timing of the first spike for each receptor type/glomerulus, with respect to the onset of respiration, as part of the response. This time is shorter for higher concentrations, and thus a linear expansion around an operating point like the one we use for firing rates could be used for timing, as well. The timing variables could be used instead of, or in addition to, the rate variables.

We now explain this in more detail in the text.

2) The authors choose an operating point where they can apply a linear model for glomerular responses. In the animal, the operating range of different receptors for different odors is rather diverse, with the half-max varying substantially and the slope also varies. Thus a subset of odors will be saturating for some receptors, but linear or even subthreshold for other receptors. I suspect that this will affect the analysis of the responses.My view is that any coding theory has to account for the very wide range ofodor concentrations encountered in nature. One could possibly add this to theanalysis reported in Equations 4 to 6, by summing the mutual information overa set of odor ranges, in which different but overlapping subsets of receptors are involved. I would be interested to see if this alters the conclusions.

There are three points here: (1) different receptors have different operating ranges for different odors, (2) odors can occur in diverse concentrations, (3) receptors have nonlinear response functions with a threshold and saturation. To address these points in a fully naturalistic setting we need precise measurements of the natural olfactory environment and a complete set of dose-response curves; comprehensive data of this kind is not available.

So, as a first step we have used available data from Hallem and Carlson, 2006 and from Saito et al.,2009. These works survey responses of a subset of receptors in fly, mouse and human to a panel of odorants. In these studies, a given receptor may respond strongly to some odorants and weakly (or not at all) to others; we use these experimentally measured receptor sensitivities. Also, in our model the threshold for informative response is effectively set by the noise level in the receptor, which was also taken from data (see Appendix 1). So, in effect, different overlapping sets of receptors respond to different odors, as the reviewer would like to see. Similarly, if the intensity of a particular odor mixture is lowered (i.e., if all the component concentrations are scaled down) then some of the receptors will stop responding informatively to some of the components of the mixture. Thus, diverse response thresholds and response gains have effectively been included in our study.

We also wanted to model olfactory environments. We are not aware of any dataset describing the actual variances and covariances that are typically observed. So, for convenience we chose a Gaussian distribution because this permits parametric variations and analytic calculations. Since we pick the odor covariance matrix randomly, some of the odorants will have a large variance and some have a small one. Thus, diverse concentration ranges have been included in our study.

Including non-linearities is more challenging because we must posit a functional form for mixture responses, and because computing mutual information with such non-linear sensing requires new computational innovations that are out of the scope of this paper. Nevertheless, we checked, using experimentally measured nonlinearities, that the broad predictions of our model will be robust. This is now reported in Appendix 3. Please also see the response above to Essential Revision #2 in the editor’s summary.

3) A major point of concern with the whole analysis is of salience. The obvious outlier here is pheromones. Enormous resources are allocated to pheromone detection, and clearly this doesn't seem to fall within the framework presented in the paper. Even with the general olfactory system, the assumption of efficient coding needs to be further mapped to the distribution of odor salience, that is, relevance for animal survival. There seems to be a subtle nod to this point in the third-last paragraph of the Discussion, where 'value of detecting different odorants' is mentioned. I feel that the point is central enough that it needs to be fully addressed. […]In summary, I think that the current paper lays out a good framework but would be much stronger if some essential ramifications of the core idea were to be addressed.

It is of course true that some odors are more meaningful to the animal than others. However, it is not clear to what extent this kind of distinction is already implemented at the level of the sensory periphery. As an example, many efficient-coding studies in early vision and audition rely on approximations in which only information transfer is taken into account, without reference to meaning or value. The predictions nevertheless yield very good agreement with experiment. This might in fact be a result of the kind of prior on broad coverage that the reviewer is suggesting – in order to achieve breadth, the system would *not* adapt to extreme variations in value. This would ensure that stimuli that are not presently valuable (but might turn out to be at some point) are not ignored in favor of the ones to which high value is currently assigned. The idea here is that filtering for value and salience should occur deeper in the brain, perhaps in the olfactory cortex which has extensive projections to and from areas associated to meaning and value. From this perspective the sensory periphery should focus on simply taking in informative signals broadly. In addition, arguments based on compressed sensing suggest that, by focusing on preserving information, the olfactory system might in fact be able to sense *any* odor that is sufficiently sparse (e.g., Krishnamurthy et al., 2017). In this case, emphasizing salience in the periphery might actually be counterproductive, leading to a *narrower* distribution of receptors than desirable. We have added some of these remarks to the Discussion.

That said, the reviewer’s suggestion of an innate, evolutionarily determined prior is very interesting. From our perspective, this prior would represent the olfactory environment that the species has been subject to over generations, and could be “deformed” relative to the statistics of actual odor occurrence to account for the special importance of some odors. Concretely, suppose a particular odorant occurs only rarely but is associated to a predator and so is disproportionately important. Then, artificially inflating the variance of that odorant in the effective background environment effectively increases its importance to the optimization. In this way, the “background” olfactory environments that we start from can be regarded as incorporating the priors suggested by the referee. In fact, the data from Ibarra-Soria et al. supports this sort of picture, showing that genetically different strains of mice have somewhat different receptor distributions even when reared in the same environment. The picture we have in mind is that the changes reported in that paper due to environmental factors can be seen as a perturbation on an innate prior which incorporates an effective long-term olfactory environment, perhaps discounted for salience, that a species has been subject to.

Finally, a fully grounded approach requires new experimental and theoretical quantification of the notion of “value”. This is a major goal of neuroscience and of the study of behavior, but the field is far from achieving this. Thus, any treatment of value in our paper can at best be a preliminary step and a detailed investigation of value lies out of the scope of this work. In effect, we are hypothesizing, like in the literature on vision and audition, that the value of signals in the early olfactory system (which does not have access to cognitive portions of the brain) is dependent largely on the information content of the signal. There may of course be some odors with a special valence, and there is some data suggesting that even the main olfactory system can adapt to these. Extensions along the lines suggested by the reviewer would incorporate such specific effects. But, since the necessary experiments are sparse at present, we believe it is best to postpone the incorporation of value until more data are available to fix the parameters.

Reviewer #2:[…] 1) The conclusions concerning how receptor abundances should change following a change in environment are disappointing. While their model recapitulates Ibarra-Soira's result which predicts that the distribution of high abundance receptors is likely to remain unchanged, they do not provide any concrete predictions on the receptors which change their abundance in either direction of change or magnitude. As currently stated, the central predictions of the model – that optimal receptor abundances can increase or decrease or stay the same following a change in environment – seems to be unfalsifiable.

As we discussed above in answer to the first point on the list of essential revisions, our model certainly makes quantitative predictions for the abundances of OSNs in the olfactory epithelium. These are of course falsifiable. Given specified changes in an olfactory environment and receptor affinities the theory makes specific predictions for which receptors will change in abundance and in which direction. We have highlighted this in the section “A framework for a quantitative test”. An important point of our paper is that the predicted changes are not readily summarized in terms of a simple catchphrase like “receptors with greater response variance should increase in number”. This is because, as we discuss in detail in the paper, in the presence of widespread correlations in the responses, the optimal abundance of one receptor depends on the context of the responses of all the others. Nevertheless, given receptor affinities and a characterization of the odor environment the model predicts changes in abundances precisely.

The doubt here may have arisen because of the way we phrased the prediction in the original text (“…receptor abundances can increase, decrease or the stay the same following a change in the environment…”). This statement may have been misread to mean that a given receptor will sometimes increase, decrease, or stay the same in replicates of the same experiment. We intended to say that receptors may increase or decrease in number after increased exposure to a particular ligand in *different* contexts, and that the effect will be reproducible, although the specific change in a receptor will depend on the context of all the others. We have edited the text throughout to state this better in order to avoid confusion. For example, we have edited the Abstract to read “Experimentally, increased exposure to odorants leads variously, but reproducibly, to increased, decreased, or unchanged abundances of different activated receptors. We demonstrate that this diversity of effects is required for efficient coding when sensors are broadly correlated, and provide an algorithm for predicting which olfactory receptors should increase or decrease in abundance following specific environmental changes.”

Incidentally, the *qualitative* predictions of our model are themselves falsifiable. It is a non-trivial observation that increasing exposure to an odorant does not necessarily lead to an increase in the abundance of the receptor types that respond to it, as we might naively expect if the lifetime of OSNs was simply tied to their activity. The fact that this counter-intuitive effect of exposure to ligands is seen experimentally is corroborative evidence for the framework that we are proposing.

The manuscript could be strengthened by making more concrete predictions about how receptor abundances should change, at least in particular regimes. For example, the authors note that for intermediate numbers of neurons, optimal receptor distributions are anti-correlated with the inverse of the overlap matrix Q^-1^. They expand on this to say that receptors with high Q-1 can be uninformative because they do not fluctuate or because they provide redundant information. Although I did not fully follow the arguments here, it seemed like this was saying that abundance is inversely related to information, and there are two ways to be uninformative, one by having low variance, and two by being highly correlated with other receptors. Could this be used to make more concrete predictions about predicted changes in receptor abundance, at least for a given number of neurons?

In some limits, the structure of the maximally informative receptor distribution can be given a simple intuitive description. We explain these limits in some detail now in the revamped section “Optimal OSN abundances are context-dependent”. However, in general the abundances are dependent on the context of all the receptor responses. For instance, in certain regimes, they are related to elements of the inverse overlap matrix, which depends on the full covariance matrix of responses. The conceptual meaning of the inverse overlap matrix in terms of response variance and predictability from other responses is now discussed in Appendix 6. To summarize again, running the optimization from Equation 7 in our model makes fully concrete predictions about receptor abundances, given the required parameters. The discussion involving the inverse overlap matrix provides an intuition explaining the results of this optimization, but it is not meant to replace it.

In addition, the authors also provide model evidence for predicting the magnitude of the change based on the change in olfactory environment, but it is unclear the characteristics which group types of changes together.

A key point of our paper is that there is *not* a simple characterization of receptors that increase *vs.* decrease in number after particular environmental changes. This is because, as we show in the paper, there is a global dependence on the context of the responses of the rest of the population of receptors. The specific, quantitative changes that should occur for a given environmental change can, however, be predicted using the full optimization framework we describe (the new section “A framework for a quantitative test”describes how to do this). We discuss this, and simple intuitions that apply in special limits (e.g., high/low SNR), in the section “Optimal OSN abundances are context dependent”.Indeed, the complex context dependence is necessary to understand the apparently sporadic patterns of change seen in experiments, and is predictable given a fuller characterization of the receptor affinities and odor environment.

To further clarify the complex patterns of change, we updated Figures 4 and 5 regarding differences in the optimal receptor distribution for different sorts of changes in the environment. Figure 4 shows results for a pair of environments that differ only in the variance of a few odorants. Figure 5 compares results for a pair of randomly differing environments and two largely non-overlapping environments. The results are discussed in sections entitled “Environmental change leads to complex patterns of OSN abundance changes” and “Changing odor identities has more extreme effects on receptor distributions than changing concentrations”.

2) Some of the conclusions seem odd when considered in the context of olfactory evolution. For example, the authors conclude that if the number of neurons is large, then the optimal receptor distribution is approximately uniform. Olfactory systems differ greatly in magnitude across organisms. In particular, two of the most-studied models, fly and mouse, differ by an order of magnitude in the number of receptors (~60 for fly, ~1000 for mouse), as well as the total number of neurons. The finding that total neuron number determines receptor distribution should be tied numerically to the olfactory systems of flies and mice, if not also for other organisms. It is unclear, for example, whether the olfactory receptor number of mice is considered large, or whether it would fall in the intermediate signal to noise regime. Does the model predict that mouse receptor distributions are uniform while fly distributions are highly skewed? Why then is any adaptation observed in mouse receptor abundances as has been observed experimentally?

We were using the terms “large” and “small” in a limiting sense, as the numbers went to infinity or one. In these limits, the effective SNR becomes either very large or very small, driving the receptor distribution to either high diversity and uniformity or low diversity and inhomogeneity (section entitled “Receptor diversity grows with OSN population size”). Such limits are useful to analyze because they give a sense of the factors and considerations that are influencing the results. The degree of diversity and inhomogeneity seen in olfactory receptor distributions in animals suggests that they are effectively in an intermediate regime between the “large” and “small” population sizes.

In the intermediate regime, the number of receptor types used and their relative abundances are determined by the interplay between receptor affinities, noise levels, and environmental odor statistics. One way of thinking about this is that, in our model, the total number of neurons *K_tot_* is a constraint, reflecting the limited resources that can be allocated to the olfactory epithelium. Given a fixed bank of receptor types, some of these will be more useful for transmitting information compared to others. Thus, not all of them will be used when the number of neurons is small, and allowing more neurons in the system allows more of the receptor types to be used (see Figures 2A, B, C). At some level of the neuron number, *K^*^_tot_*, all receptor types will be used. We are definitively in the large-neuron-number regime if *K_tot_* is much larger than *K^*^_tot_*. Since this is effectively defined by using all the receptor types available, it makes sense that it increases with the size of the receptor bank.

Our assumption is that the biologically-relevant regime is typically one where *K_tot_* is intermediate, comparable to *K^*^_tot_*. This is because for smaller numbers of neurons, some receptor types would not be used, and thus we would expect these to mutate into non-functional forms. And for much larger numbers of neurons, the improvement in information transmission would no longer be significant. Thus, we would expect that the OSN population size will be selected over time such that the functional receptors are all useful, and such that there is not much information benefit to having more neurons. This is the “intermediate” regime of our analysis, and both fly and mouse should be in it. We would thus expect the number of receptor types in mouse to be larger than in fly, given the increased size of the epithelium. If the receptor pool that both animals used was the same, and if the odor environments they experienced were the same, then Figure 2C would provide a quantitative prediction for exactly how their number of receptor types and number of OSNs are related. Of course, insect olfactory receptors are evolutionarily distinct from mammalian ones, and the environments that flies and mice inhabit may have very different odor statistics. Thus, we cannot directly compare their receptor repertoires although it is indeed true that mice have more OSNs and more receptor types. Such a comparison can perhaps more meaningfully be done for different species of mammals, and we now include some results in Figure 2F.

Given the results presented here one might imagine that the optimal strategy would be to make a very large number of broadly tuned receptors. Instead, what we observe across evolution are olfactory systems of various sizes, with various widths of odor tuning, all constantly evolving.

As we explained in answer to point 3) of the editor's summary, we feel that our goals in relation to the evolution of olfactory systems have been misunderstood. In our model, the tuning of olfactory receptors is taken as given. Our model does not say anything about what the tuning should be. In any case, it could be that the characteristics of receptors depend on biochemical properties that do not allow them to all be similarly broadly-tuned, even if it turned out that this was optimal from an information-transmission viewpoint. Actually, it is not even obvious that broad tuning is necessary given the presumed combinatorial nature of the odor code. For example, suppose any given odor elicits responses in just 10 out of 100 receptors. There are O(10^13^) such patterns, more than enough to encode the possible species of volatile molecules an animal is likely to encounter. That said, given a certain number of noisy receptor types, it will still be useful to re-distribute them to best represent the particular odor scenes that an animal encounters. It is also worth noting that the notion of tuning width is always dependent on (a) which odorants we test the receptor with, and (b) an arbitrary threshold separating what we call an active receptor vs. an inactive one. Thus, the same data can look broadly-tuned to some researchers and sparse to others. Studies that attempt to answer the question of how an optimal olfactory receptor repertoire should be built exist (e.g., Zwicker et al. 2016), but we stress again that their goals are complementary to ours.

The number of receptors in particular seems to be under strong evolutionary pressure, with new gene families expanding (as in ant ORs) or collapsing (as in humans). This discrepancy, or the other constraints that might lead to the biological situation, should be commented on.

Again, we do not attempt to model the evolution of olfactory receptor genes, or to find an optimal set of receptor types. For work along these lines, see Zwicker et al., 2016. Our model instead focuses on a complementary question that takes the available receptor types for granted and asks how these should be used, i.e., how many receptors of each type should an animal have. For a related perspective in early vision see Ratliff et al., 2010, where the tuning of ON and OFF cells in the retina is assumed, and the relative fractions of these types is predicted. That said, there is a possible connection with the collapse of the OR gene family in humans, in that the optimal receptor repertoire for the typical odor environment of interest to humans might have included vanishing or negligible amounts of some of the available receptor types. If this were the case, we would expect these ORs to mutate to non-functional forms due to genetic drift. To test this hypothesis, we would need a good grasp on the way in which human environments and olfactory behaviors differ from those of our remote ancestors. We do not have such data, but new Figure 2F and the associated discussion in the section “Increasing OSN population size” bear broadly upon these points.

The authors state that receptor abundances do not change in insects and therefore focus on a mammalian example to test their hypothesis. However, insect olfactory systems evolve quite rapidly between closely related species, and there is a large literature on this, especially from the Hansson group (e.g. Dekker…Hansson, 2006). Can these studies be used to test any of the hypotheses here?

The prediction of the way in which the number of receptor types grows with the number of OSNs is contingent on the receptor repertoire and the environment being similar between the species we are comparing. This makes the rapid evolution of insect olfactory systems a hurdle, rather than an advantage, for detailed comparison. For instance, in Dekker et al., 2006, the generalist *D. melanogaster* is compared to the highly specialized *D. sechellia.* It is clear that the typical olfactory environments for the two species are very different, and it would thus be difficult to say to what extent our prediction should hold without having measurements of these environments.

We have added a plot (Figure 2F) showing how the number of intact OR genes scales with a measure of the size of the olfactory epithelium across several species of mammals. While the trend in these data are in agreement with our model, we stress that there are many caveats about this comparison, as described in the text.

Insects could perhaps be used in experiments in which the olfactory environment is tightly controlled. This would be interesting to do, and our model could be tested in this context.

Or can the authors propose comparative studies that would test their hypotheses?

Studies very similar to that in Ibarra-Soria et al., where mice were raised in two different olfactory environments, would be ideal for testing our model. This would involve measuring the statistics of a few dozen odorants in the environment of control mice, and the same statistics for the exposed group, combined with the response profiles of a set of mouse ORs to those same odorants. Using these parameters, our model would give precise quantitative predictions (including signs) regarding the amounts by which the abundances of different ORs should change. Given the approximations we make, we would not expect these to be in exact agreement with the experimental values, but we would expect a significant correlation. This would be a strong test of our hypothesis. We now explain this in, e.g., the section “A framework for a quantitative test”.

3) Several concepts used in the text are a bit unclear, at least to a biological reader:Could the authors provide some intuition for what is meant (biologically) by the inverse of the overlap matrix?

Thank you for this question. Interestingly, the elements of the inverse overlap matrix characterize how much the responses of one receptor type can be predicted if we know the responses of the others. This predictability might happen for various reasons – for instance, some receptor responses might not vary much, and then they can be easily predicted. Note that this depends on the environmental statistics of odors – receptors that do not vary much in one environment might well vary a lot in another environment. Another reason for a receptor response to be predictable would be if its affinity profile is similar to that of other receptors. Finally, it could be that, due to properties of the odor environment, certain odorants that activate receptor *a* are always accompanied by odorants that activate receptor *b*; in this case, one receptor type's response would be predictable given the others, even though their affinity profiles could be completely different.

To be a little more precise, the off-diagonal elements of the inverse overlap matrix, *A_ab_* are related to the correlation coefficients between the responses in two glomeruli, *a* and *b*, while controlling for the responses of all the others. They are also inversely proportional to the product of the standard deviations of the two responses. The diagonal elements *A_aa_* depend inversely on the variance of the response of the *a*^th^ glomerulus, and are also related to a parameter which measures how well the response in the *a*^th^ glomerulus can be linearly predicted from responses in all the others. In this way the correlation of optimal receptor distribution with the inverse overlap matrix has an intuitive interpretation: receptors which either do not fluctuate much or whose values can be guessed based on the responses of other receptors should have low abundances. We now state this at the end of the section “Optimal OSN abundances are context dependent” and develop the details further in Appendix 6.

Could the authors please unpack the following sentence:The quantity KQ thus behaves as a signal-to-noise ratio (SNR), so that Equation 4 is essentially a generalization to multiple, correlated channels of the standard result for a single Gaussian channel, I = 1 log(1 + SNR^2^).

This sentence was indeed difficult to follow in part because noise in the receptors had been absorbed into various expressions as an effective normalization. To increase clarity, we decided to reinstate the noise standard deviations for each receptor type, Equation 2, so that both responses and concentrations can be measured in natural units (e.g., firing rate and molarity, respectively). Now the quantity that used to be *KQ* is seen to actually be *K**𝛴*^-1^*Q*, which is, in matrix form, the ratio between the covariance matrix of glomerular responses (*Q*) and the covariance matrix for the noise (𝛴 *K^-1^*), where the *K^-1^* term corresponds to the decrease in noise variance due to averaging over OSNs with the same receptor. When glomerular responses are uncorrelated (i.e.*, Q* is diagonal), the determinant in Equation 4 is easily calculated, and we obtain *I* = sum over all receptor types of 1/2*log(1 + SNR*_i_*^2^), where SNR*_i_* is the signal-to-noise ratio in channel *i*, SNR*_i_*^2^ = *Q_ii_* / (𝜎*_i_*^2^ / *K_i_*). The result 1/2*log(1+SNR^2^) for the mutual information in a Gaussian channel has been known since the work of Shannon (1948), and so we wanted to emphasize the connection to this classic result. We have tried to clarify these points in the section “Information Maximization”. More technical details are also presented in Appendix 2.

Could the authors please clarify in the discussion of Equation 7 whether K_tot_ represents the total number of neurons, the number of receptors, or the number of receptor types?

*K_tot_* refers to the total number of neurons. We updated the text.

Is the total number of neurons the most sensible thing to vary or would it be interesting to look at olfactory systems with different numbers of receptor types? This seems related to the question of where noise arises in the system, and what other constraints, besides information as quantified here, an animal might have on the design of its olfactory system.

Our model takes the odorant affinities for each receptor as an input. This means that in order to increase the number of receptor types, we need to fix the affinities of the added receptors. There are many ambiguities in doing this. Decreasing the number of receptor types is, however, more straightforward – we can simply remove some receptors from the analysis. We now do this in Figure 7. This analysis has an auxiliary benefit: it provides a test of how robust the receptor distribution is to changes in the repertoire of available receptor types.

4) The investigation of how optimal coding changes with broad versus narrow tuned receptors was interesting. However, real receptor arrays, at least as seen in the Hallem data, contain a mix of broadly and narrowly-tuned receptors, and receptor tuning width depends on odor intensity, with many receptor showing narrowly tuned response at low concentrations and wider tuning at high concentrations. Could the authors explore what happens in this regime, and provide any explanation for why animals might have both broad and narrowly tuned receptors? This finding could be further explored by making predictions for olfactory systems with receptors of mixed tuning widths, as is generally accepted to be the case in most organisms. This would provide a more concrete prediction for future experiments.

In various of our results (Figure 2A, B, C, Figures 3, 6 and 7) we are indeed using data from Hallem, Carlson, 2006 and also from Saito et al., 2009. These data include both broadly and narrowly tuned receptors. In the original submission we had also examined situations where all the tuning widths were narrow or wide (Figure 2D, E). We now also present additional results where the artificial receptor arrays are made up of heterogeneous receptors with varying tuning widths (Figure 5).

Regarding the question of why animals have both broadly-tuned and narrowly-tuned receptors, we stress again that in our study the affinity profile of the available receptor genes is considered as given. As such, our model is not trying to address the optimal way to build the receptor repertoire, but simply the optimal way of using this repertoire (i.e., relative proportions of different receptor types).

We also emphasize again that our model makes fully precise, quantitative predictions once the affinity profile for olfactory receptors is known. Thus, if we are interested in making testable predictions, the best approach is to use measured affinity profiles. In this case, we do not need to worry about how to choose the tuning widths for the receptors since nature has already chosen them for us.

5) The authors claim that their model is robust to non-linearities and as well as their choice to represent the olfactory environment as a vector of concentrations. These ideas should be tested and demonstrated within the paper.

We have made these points more precise in the main text and added two sections in the Appendices to explicitly address them. The section entitled “Invariance of mutual information under invertible and differentiable transformation”in Appendix 2, is a mathematical explanation of the statement in the section title. For example, consider a linear-nonlinear model,

*r_a_ = g_a_(K_a_ S_ai_ c_i_ + 1/sqrt{K_a_}* 𝜂*_a_),* with *g_a_* a set of invertible functions. It is a mathematical identity that the mutual information is invariant under such transformations. Of course, some nonlinear transformations may not preserve information if they do not satisfy the stated conditions. From a biological standpoint, it is most interesting to study nonlinearities like those in competitive binding models like the ones suggested by the reviewer. We have now examined these in Appendix 3,as explained in the response to the next question, as well as in answer to point 2) from the editor's summary.

Regarding the way in which the olfactory environment is represented, we were trying to say that as far as our model is concerned, it does not matter what the numbers *c_i_* represent: the same kind of model with the same generic mathematical results would apply if *c_i_* were concentrations of distinct odorants or if they were, for instance, aggregates over several chemical species related by some property. Of course, the parameters feeding into the model – the sensing matrix *S* and the environment covariant matrix 𝛤 – depend on the meaning of the environment vector *c_i_*, and thus the specific results would change. However, the analysis itself wouldn't. We have tried to make this clearer in the text in the section “Olfactory response model”.

For example, the nonlinearities involved in receptor encoding are well known: receptor responses can be expressed as a Hill function of odor concentration:r = (c^n)/(c^n+Kd)In many olfactory systems n=1, further simplifying this equation. The authors should explicitly show that the model generalizes when this nonlinearity is included.

To our knowledge, there is still debate regarding models for how olfactory receptors respond to odors. While the response to single odorants can be well-approximated by Hill functions, the response to mixtures is harder to describe. Competitive binding models do perform relatively well (Singh et al., 2018), as in fact do linear approximations for small numbers of mixture components and in the regime between the response threshold and saturation. We used a linear approximation in the main text because it uses fewer parameters and is analytically tractable. Furthermore, a complete competitive binding model of the sort reported in Singh et al., 2018 requires measurements of dose-response curves of all the receptors being studied against all odorants of interest. We do not have such data. Therefore, we used existing data to create a simple competitive binding model for a few receptors and compared the results (obtained through a numerical analysis) with those obtained analytically from the linear approximation. The new Appendix 3 shows that the results are broadly similar between the two models. As we explain in more detail above in the response to point 2) in the Essential revisions, our linear sensing model provides a reasonable and tractable approximation which can be numerically extended to a fully nonlinear model when such data become broadly available for more OSNs.

In addition, the main sources of noise in receptor encoding are likely to be (1) difference in receptor abundance across neurons that express the same receptor, (2) stochasticity in receptor binding and activation. The authors might consider incorporating these sources noise and showing that the model extends in this case.

We agree that these separate sources of noise may be present in the olfactory periphery, but we are not aware of specific, quantitative noise models for which the relevant parameters have been measured. As above, we think that in the absence of these data, it makes more sense to start with a simple, analytically-tractable noise model, and to leave more complex descriptions for future work when experimental guidance is available. Also, the qualitative structure of our results is not going to depend on the source of noise. For example, even after including the two sources of independent noise the referee mentions separately, the optimal distribution of receptors is still going to be context-dependent.

The first section of the Results is difficult to read because it contained a number of statements justifying elements of the model and claiming that these do not affect the conclusions. This section would be easier to read if these points were saved for later in the manuscript where they could be explicitly demonstrated.

We went through the section and tried to improve the presentation in the manner suggested by the reviewer.

6) The section on dynamical optimization at the end seemed least well-constrained by data, and also (as noted) somewhat preliminary. The authors might consider reserving this material for a future manuscript that explores dynamics and tests them more thoroughly.

The dynamical model part of the paper is intended as an indication that simple birth and death processes modulated by experience can achieve the sorts of optima that our model describes. We feel that it is useful to see this, and have retained the section.

Efficient-coding arguments of the sort that we used in the paper are normative: we derived optimal rules for how the olfactory periphery should be organized, given a simplified model of receptor responses, and we argued that organisms would benefit from approaching these optima. But a fundamental issue with such approaches is that there is no guarantee that the optimum is actually reachable using the resources available to an organism. This is why we believe it is important to show that the optima can be dynamically reached. It is interesting that our dynamical model requires that the death rate of the neurons, but not their birth rate, should depend on olfactory experience, as experimentally observed.

And instead using this space to show that the model still holds when certain assumptions in the first version of the model are relaxed.

We have made extensive additions to paper, both in the main text and in the Appendices to relax various assumptions. Nonlinearities, diversity of tuning widths, realistic receptor affinity distributions, changing the number of receptor types, and changing the tuning widths are all now addressed as described above.

7) The authors should consider including graphical representations, similar to those provided in Figure 1, for concepts such as the mutual information measure, the covariance matrix, the overlap matrix, and the inverse overlap matrix. This would help provide insight for readers with less mathematical background, who may nonetheless be interested in the predictions of the models.

We have revamped all of our figures to aid the reader. We decided not to give an introduction to information theory (e.g., the notions of mutual information, covariance matrix) because there are many standard textbooks and also review articles in the field of neuroscience itself. However, we did include a more extensive discussion of the overlap matrix, as described above. In addition, we have now included a section in the paper (*A framework for a quantitative* test) and a Matlab script that makes it easy to plug measurements of receptor affinities and natural odor statistics into our model and obtain the predicted optimal OSN numbers. This allows people who are not expert in the mathematics behind the model to still use it or test it.

Reviewer #3:[…] The manuscript is well written and the arguments are clearly presented for the most part. My main concerns with the manuscript are that some limitations are not spelled out explicitly and that the theoretical analysis could have been more comprehensive. In particular, the authors do not investigate how their model would fair in the realistic case where odors are sparse and they do not discuss how the results depend on the number of different receptor types and the number of different odor molecules.

We have added an analysis of the latter: Figure 7 now shows the amount by which the results change when receptors and/or odorants are subsampled. This also allows us to see that our overall results are robust to such subsampling.

Natural odors like foods are often sparse, containing maybe 40-50 components that are important for perception. Recent work has suggested that this sparsity may influence the structure of the sensing matrix implemented by the receptor repertoire (e.g., Zwicker et al., 2016, Krishnamurthy et al., 2017). However, an odor environment typically contains a mixture of such odors, and many odorants can activate multiple receptors. In this context it is unlikely that the odor response is sparse even if many individual natural odors only contain a few tens of components. The correlated structure of the resulting responses is the key factor driving our results. Thus, we should expect the results to remain qualitatively the same for environments consisting of combinations of odors that are sparse in chemical space.

An attempt to directly model odor sparsity runs into the same lack of data about the structure of natural odor environments that we discussed in our response to the overall Essential revisions. We have some idea that individual natural odors (e.g., strawberry) typically contain a few tens of components (perhaps 40-50) but we do not have a detailed survey over odors. We also do not know how many of these odors co-occur in natural settings and with what frequencies, variances and co-variances. Right now we are in a setting where knowing even just the mean and covariance matrix of a set of odorant concentrations would be an important advance. In this context, modeling olfactory environments requires exploring many arbitrary choices each of which would require separate justification. We could vary over all of these choices, but that would require a study in itself. In fact, the random covariance matrices that we generate have a structure reminiscent of sparse odors when the 𝛽 parameter is small (see Appendix 4 for a discussion of how these matrices are generated) and are thus a good starting model of a sparse odor environment (i.e., where each odor is composed of a small fraction of all odorants). The next step might be to treat the odor environment as a mixture of such Gaussians in which each term of the mixture represents an odor object, itself a sparse odor. But many choices are necessary – how many odor objects to include, how many odorants are present in each one, variance and covariance of the objects, etc. In addition, calculating mutual information from such distributions is numerically challenging if we include many odors and odorants as necessary to be realistic (this is discussed further above and in Appendix 3). Meeting that challenge would be worthwhile if we knew enough about the odor environment to make it useful, and the effort would make an interesting computational paper in its own right, but is out of the scope of the present work.